# Systems genetics uncover new loci containing functional gene candidates in *Mycobacterium tuberculosis*-infected Diversity Outbred mice

**Daniel M. Gatti**[1], **Anna L. Tyler**[1], **J Matthew Mahoney**[1], **Gary A. Churchill**[1], **Bulent Yener**[2], **Deniz Koyuncu**[2], **Metin N. Gurcan**[3], **MK Khalid Niazi**[3], **Thomas Tavolara**[3], **Adam Gower**[4], **Denise Dayao**[5], **Emily McGlone**[5], **Melanie L. Ginese**[5], **Aubrey Specht**[5], **Anas Alsharaydeh**[6], **Philipe A. Tessier**[7], **Sherry L. Kurtz**[8], **Karen L. Elkins**[8], **Igor Kramnik**[9], **Gillian Beamer**[6]*

**1** The Jackson Laboratory, Bar Harbor, Maine, United States of America, **2** Rensselaer Polytechnic Institute, Troy, New York, United States of America, **3** Wake Forest University School of Medicine, Winston Salem, North Carolina, United States of America, **4** Clinical and Translational Science Institute, Boston University, Boston, Massachusetts, United States of America, **5** Tufts University Cummings School of Veterinary Medicine, North Grafton, Massachusetts, United States of America, **6** Texas Biomedical Research Institute, San Antonio, Texas, United States of America, **7** Department of Microbiology and Immunology, Laval University School of Medicine, Quebec, Canada, **8** Center for Biologics Evaluation and Research, Food and Drug Administration, Silver Spring, Maryland, United States of America, **9** National Emerging Infectious Diseases Laboratories, Boston University, Boston, Massachusetts, United States of America

* GBeamer@txbiomed.org

**Data Availability Statement:** Data has been deposited in Gene Expression Omnibus (GEO), and assigned Series ID GSE179417.

## Abstract

*Mycobacterium tuberculosis* infects two billion people across the globe, and results in 8–9 million new tuberculosis (TB) cases and 1–1.5 million deaths each year. Most patients have no known genetic basis that predisposes them to disease. Here, we investigate the complex genetic basis of pulmonary TB by modelling human genetic diversity with the Diversity Outbred mouse population. When infected with *M. tuberculosis*, one-third develop early onset, rapidly progressive, necrotizing granulomas and succumb within 60 days. The remaining develop non-necrotizing granulomas and survive longer than 60 days. Genetic mapping using immune and inflammatory mediators; and clinical, microbiological, and granuloma correlates of disease identified five new loci on mouse chromosomes 1, 2, 4, 16; and three known loci on chromosomes 3 and 17. Further, multiple positively correlated traits shared loci on chromosomes 1, 16, and 17 and had similar patterns of allele effects, suggesting these loci contain critical genetic regulators of inflammatory responses to *M. tuberculosis*. To narrow the list of candidate genes, we used a machine learning strategy that integrated gene expression signatures from lungs of *M. tuberculosis*-infected Diversity Outbred mice with gene interaction networks to generate scores representing functional relationships. The scores were used to rank candidates for each mapped trait, resulting in 11 candidate genes: *Ncf2*, *Fam20b*, *S100a8*, *S100a9*, *Itgb5*, *Fstl1*, *Zbtb20*, *Ddr1*, *Ier3*, *Vegfa*, and *Zfp318*. Although all candidates have roles in infection, inflammation, cell migration, extracellular matrix remodeling, or intracellular signaling, and all contain single nucleotide polymorphisms (SNPs), SNPs in only four genes *(S100a8, Itgb5, Fstl1, Zfp318)* are predicted to have deleterious effects on protein functions. We performed methodological and candidate validations

**Funding:** • The work was supported by funding from the National Institutes of Health R21 AI115038 (GB); the National Institutes of Health R01 HL145411 (GB); the American Lung Association Biomedical Research Grant RG-349504 (GB); Tufts University's Cummings School of Veterinary Medicine Summer Research Program (AS), and the National Institutes of Health HL126066 (IK). The funders had no role in study design, data collection and analysis, decision to publish, or preparation of the manuscript.

**Competing interests:** The authors have declared that no competing interests exist.

to (i) assess biological relevance of predicted allele effects by showing that Diversity Outbred mice carrying PWK/PhJ alleles at the H-2 locus on chromosome 17 QTL have shorter survival; (ii) confirm accuracy of predicted allele effects by quantifying S100A8 protein in inbred founder strains; and (iii) infection of C57BL/6 mice deficient for the *S100a8* gene. Overall, this body of work demonstrates that systems genetics using Diversity Outbred mice can identify new (and known) QTLs and functionally relevant gene candidates that may be major regulators of complex host-pathogens interactions contributing to granuloma necrosis and acute inflammation in pulmonary TB.

## Author summary

We investigated the genetic basis of susceptibility to *Mycobacterium tuberculosis* using Diversity Outbred mice, a mouse population suited for studying complex genotype-phenotype relationships. We found five new genetic loci and three known loci. Three loci associated with multiple correlated disease traits likely contain genes that are major regulators of host inflammatory responses which favor *M. tuberculosis* growth. Further, these three loci contain four gene candidates with single nucleotide polymorphisms that are predicted to have deleterious effects upon protein functions.

## Introduction

The number of humans who develop active pulmonary tuberculosis (TB) is minor compared to those who eliminate or control *Mycobacterium tuberculosis* (5–10% *vs* 90–95%), yet morbidity and mortality from TB remain high. COVID-19 mortality temporarily surpassed global TB mortality, but TB has remained in the top two leading causes of death due to an infectious disease for decades, killing more people than HIV/AIDS and malaria. Pulmonary TB is the most common and most contagious form of TB, with mortality rates >40% if untreated and higher if caused by antibiotic resistant *M. tuberculosis* [1–8]. Human responses to *M. tuberculosis* range from fulminant pulmonary TB that develops within weeks to lifelong control or complete clearance of bacilli [9–11]. Further, a body of evidence shows an interesting paradox: Immune competence is necessary to restrict *M. tuberculosis* growth [12], but is not sufficient to prevent disease [13].

The variable responses to *M. tuberculosis* and lack of single gene defects in most patients indicate a complex genetic basis for pulmonary TB, and this has been investigated by linkage association mapping, genome wide association studies, and other methods, recently reviewed [14–17]. Reviews identify knowledge gaps attributable to using laboratory mouse strains that do not replicate key disease traits of human pulmonary TB, e.g., granuloma necrosis [18–23]. To address these gaps, we and others investigate genetic contribution, molecular pathogenesis, and vaccine efficacy in the Diversity Outbred mouse population, panels of Collaborative Cross inbred mice and their progeny, some of which do develop human-like pulmonary TB [24–29]. These mice are valuable resources to model complex genotype-phenotype associations; tools to dissect the genetic basis of disease; and a means to assess effects of candidate genetic polymorphisms *in vivo*.

The Diversity Outbred mouse population originated by breeding eight inbred founder strains together, resulting in a reproducible experimental population with balanced allele frequencies of one-eighth across the genome [30]. This is important for genetic mapping studies

because low allele frequencies in natural populations can diminish power and increase false positive associations [31]. Further, Diversity Outbred mice carry over 40 million variants [32], some of which alter regulatory elements, splice sites, and protein-coding sequences. This defined genetic architecture allows rigorous investigation of genotype-phenotype association in context of *M. tuberculosis* infection. Here, we selected phenotypes of pulmonary TB (i.e., quantified traits) for biological relevance, many of which are shared by humans, non-human primates, and Diversity Outbred mice [33]. Traits were selected to span clinical, granuloma, microbiological, and immune/inflammatory mediators that are known to positively correlate with disease, or are investigated as diagnostic biomarkers, or have been shown to activate or negatively regulate immune/inflammatory responses to *M. tuberculosis*. Briefly, the clinical trait of disease was weight loss. The granuloma trait was necrosis. The microbiological trait was *M. tuberculosis* lung burden. Chemokine correlates of disease were CXCL1, CXCL2, CXCL5 [24,34]. TH1 proinflammatory cytokines and acquired immune mediators were tumor necrosis factor (TNF) [35], interferon-gamma (IFN-gamma) [36], and interleukin (IL)-12 [37]. Biomarker candidates were S100A8, matrix metalloproteinase (MMP)8, and vascular endothelial growth factor (VEGF) [34] and the immune regulatory cytokine was IL-10 [38].

To find genetic loci associated with pulmonary TB traits, we used quantitative trait locus (QTL) mapping. Next we ranked candidate genes within the *Diversity Outbred tuberculosis susceptibility* (*Dots*) loci that were associated with correlated, colocalized traits by using a machine learning algorithm [39,40] to find genes functionally related to the mapped traits and the fit models scored each candidate [41]. All candidates contain a variety of SNPs as annotated in Mouse Variation Registry (MVAR). Seven of the eleven candidates contain missense SNPs in protein coding regions, and of those, the SNPs in four candidates (*S100a8*, *Itgb5*, *Fstl1*, *and Zfp318)* are predicted to have deleterious consequences on protein functions. Published *in vitro* and *in vivo* studies show roles for three candidates (*Itgb5*, *Fstl1*, *S100a8*) in bacterial lung infections including *M. tuberculosis* [34,42–47], although questions remain. To our knowledge, the other eight candidates have no published roles in *M. tuberculosis* infection but have been shown in other systems to contribute to cell stress responses, signaling pathways, adhesion and migration; extracellular matrix synthesis, tissue remodeling and angiogenesis; immune cell metabolism; macrophage inflammatory responses; and viral hepatitis [48–60]. Overall, ten candidate genes have roles in innate immune responses suggesting that genetically controlled responses of epithelial and endothelial cells, neutrophils, and monocytes, macrophages to *M. tuberculosis* bacilli are the primary drivers of susceptibility to *M. tuberculosis* and to disease progression in pulmonary TB. Only one candidate (*Zfp318*) has a direct role in acquired, antigen-specific immunity.

## Methods

### Ethics statement

Tufts University's Institutional Animal Care and Use Committee (IACUC) approved this work under protocols G2012-53; G2015-33; G2018-33; and G2020-121. Tufts University's Institutional Biosafety Committee (IBC) approved this work under registrations: GRIA04; GRIA10; GRIA17, and 2020-G61.

### Mice

Female Diversity Outbred mice from generations 15 16, 21, 22, 34, 35, 37 and 42 and the inbred founder strains: A/J, C57BL/6J, 129S1/SvlmJ), NOD/LtJ, NZO/HILtJ, CAST/EiJ, PWK/PhJ, and WSB/EiJ mice were purchased from The Jackson Laboratory (Bar Harbor, ME) and group housed (n = 5–7 mice per cage) on Innovive (San Diego, CA) or Allentown Inc

(Allentown, NJ) ventilated, HEPA-filtered racks in the New England Regional Biosafety Laboratory (Tufts University, Cummings School of Veterinary Medicine, North Grafton, MA) or at The Ohio State University, Columbus, OH. The light cycle was 12 hours of light; 12 hours of dark. Two breeding pairs of female and male C57BL/6 inbred mice carrying null mutation for *S100a8* gene were a gift from Dr. Philippe Tessier, Department of Microbiology and Immunology, Faculty of Medicine, Université Laval. After quarantine, breeders were used to establish a colony of S100a8 homozygous knock out (KO), heterozygous (HET) and wild-type (WT) C57BL/6 inbred mice, and genotypes confirmed (Transnetyx, Cordova, TN). Mice were housed in disposable sterile caging or re-usable autoclaved caging containing sterile corn-cob bedding, with sterile paper nestlets (Scotts Pharma Solutions, Marlborough, MA), and/or sterile enrichment paperboard or plastic "houses." Cages were changed every other week or sooner if soiled. Mice were provided with sterile mouse chow (Envigo, Indianapolis, IA) and sterile, acidified water *ad libidum*.

## *M. tuberculosis* Aerosol Infection

Female Diversity Outbred mice and inbred founder strains were infected with aerosolized *M. tuberculosis* strain Erdman bacilli using a custom-built CH Technologies system [24,34,61] or a Glas-col (Terre Haute, ID) system [62,63] between eight and ten weeks of age. Male and female C57BL/6 *S100a8* KO, HET, and WT mice were infected between eight and sixteen weeks of age. For each aerosol infection, the retained lung dose was determined by euthanizing a cohort of four to twelve mice 24 hours after exposure, homogenizing the entire lungs in 5mL sterile phosphate buffered saline, and plating the entire homogenate onto OADC-supplemented 7H11 agar. After 3–4 weeks at 37˚C, *M. tuberculosis* colony forming units were counted. Mice were infected with ~100 colony forming units in the first two experiments, and ~25 colony forming units in the subsequent eight experiments.

## Quantification of TB-related Traits (Phenotyping)

*Survival*. IACUC protocols disallowed natural death as an endpoint. Therefore, as a proxy of survival, we used the day of euthanasia due to any single criterion: Severe weakness/lethargy; respiratory distress; or body condition score < 2 [64]. We confirmed morbidity was due to pulmonary TB by finding: (i) Large nodular, or severe diffuse lung lesions; (ii) histopathology confirmation of severe granulomatous lung infiltrates; (iii) growth of viable *M. tuberculosis* colonies from lung tissue; and (iv) absence of other diseases. Twenty-one *M. tuberculosis* infected Diversity Outbred mice were excluded due to co-morbidity that developed during the in-life portions.

*Weight loss*. Mice were weighed 1 to 3 days prior to *M. tuberculosis* aerosol infection, at least once per week during infection, and immediately before euthanasia. For each mouse, weight loss was calculated as the percent loss from peak body weight.

*Lung granuloma necrosis*. Immediately after euthanasia, lung lobes were removed and inflated and fixed in 10% neutral buffered formalin (5–10 mL per lobe), processed, and embedded in paraffin, sectioned at 5μm, and stained with hematoxylin and eosin with or without carbol fuschin for acid-fast bacilli at Tufts University, Cummings School of Veterinary Medicine, Core Histology Laboratory (North Grafton, MA). Hematoxylin and eosin-stained glass slides were magnified 400 times and digitally scanned by Aperio, LLC (Sausalito, CA) ScanScope scanners at 0.23 microns per pixel at The Ohio State University's Comparative Pathology and Mouse Phenotyping Shared Resources Core resource (Columbus, OH) or by Aperio, LLC (Sausalito, CA) AT2 scanners at 0.23 microns per pixel at Vanderbilt University Medical Center's Digital Histology Shared Resource (Nashville, TN). Lung granuloma necrosis was

quantified in one lung lobe per mouse by our previously validated, deep learning image analysis method [65] and reported here as a ratio of granuloma necrosis per lung tissue section area.

*M. tuberculosis lung burden.* Immediately after euthanasia, 2 or 3 lung lobes were removed from each mouse and homogenized in sterile phosphate buffered saline (1mL per lobe), serially diluted, plated onto OADC-supplemented 7H11 agar, incubated at 37˚C for 3–4 weeks, after which colonies were counted, and *M. tuberculosis* lung burden in the lungs was calculated as described [66].

*Lung cytokines and chemokines.* Lung homogenates were stored at -80˚C until the experiment ended. Lung homogenates were then thawed overnight at 4˚C, serially diluted and tested for CXCL5, CXCL2, CXCL1, tumor necrosis factor (TNF), matrix metalloproteinase 8 (MMP8), S100A8, interferon-gamma (IFN-γ), interleukin (IL)-12p40, I-L12p70, IL-10, and vascular endothelial growth factor (VEGF) by sandwich ELISA using antibody pairs and standards from R&D Systems (Minneapolis, MN), Invitrogen (Carlsbad, CA), eBioscience (San Diego, CA), or BD Biosciences (San Jose, CA, USA), per kit instructions. Lung homogenate ELISA results from five of the experiments using Diversity Outbred mice have been published and analyzed for biomarkers previously [34].

## Phenotype correlation

We took the log of each phenotype after adding one (to ensure that zero was not converted to negative infinity) and regressed out the effect of the experimental batch. We then standardized the residuals and estimated the Pearson correlation between all pairs of phenotypes.

## Gene expression

Gene expression profiling was performed on a subset of 117 Diversity Outbred mice representing the spectrum of observed phenotypes. Detailed selection description, additional analyses, and released datasets are described in [34,67]. Briefly, one lung lobe was homogenized in TRIzol, stored at -80˚C, and RNA was extracted using Pure Link mini-kits (Life Technologies, Carlsbad, CA). Boston University's Microarray and Sequencing Resource Core Facility (Boston, MA) confirmed quality and quantity were sufficient for microarray analyses. Mouse Gene 2.0 ST CEL files were normalized to produce gene-level expression values using the implementation of the Robust Multiarray Average (RMA) in the affy package (version 1.62.0) included in the Bioconductor software suite and an Entrez Gene-specific probeset mapping (24.0.0) from the Molecular and Behavioral Neuroscience Institute (Brainarray) at the University of Michigan. Array quality was assessed by computing Relative Log Expression (RLE) and Normalized Unscaled Standard Error (NUSE) using the affyPLM package (version 1.59.0). The CEL files were also normalized using Expression Console (build 1.4.1.46) and the default probesets defined by Affymetrix to assess array quality using an AUC metric computed from sets of negative and positive control probes; all samples used in this analysis had an AUC > 0.8. To remove microarray probes that intersected with Diversity Outbred SNPs, we intersected the Diversity Outbred founder strain SNPs [68] with the vendor-provided probes and removed probes containing SNPs. All microarray analyses were performed using the R environment for statistical computing (version 3.6.0). A related microarray dataset and other secondary analyses have been published elsewhere [34,67] and deposited in Gene Expression Omnibus (GEO), and assigned Series ID GSE179417.

## Genotyping

We collected tail tips from each Diversity Outbred mouse and sent them to Neogen (Lincoln, NE) for genomic DNA isolation and genotyping. Neogen genotyped the mice on the Illumina

GigaMUGA platform, which contains 143,259 markers [69]. Genotypes of *S100a8* KO, HET, and WT C57BL/6 inbred mice were confirmed by polymerase chain reaction (TransnetYX, Cordova, TN).

## Haplotype reconstruction and SNP imputation

We used 137,302 GigaMUGA marker positions located on the autosomes and chromosome X found at https://github.com/kbroman/MUGAarrays/blob/main/UWisc/gm_uwisc_v1.csv and the R package *qtl2* to reconstruct the Diversity Outbred haplotypes using the founder and Diversity Outbred allele calls, and used the haplotype reconstructions to impute the founder SNPs onto the Diversity Outbred genomes [70].

## Quantitative Trait Locus (QTL) mapping

We performed multiple rounds of genetic mapping to identify the least noisy dataset, ensure appropriate sample size for statistical power, and generate robust results. The final QTL mapping results include all available genotype and phenotype data from Diversity Outbred mice that survived *M. tuberculosis* infection for up to 250 days because this dataset produced the most statistically rigorous results, and minimized confounding effects of age-related comorbidities (e.g., lung tumors, lymphoma, uterine lesions). We used *qtl2* [70] to perform linkage mapping using the founder haplotypes and association mapping using the imputed SNPs. We calculated the kinship between mice using the leave-one-chromosome-out method, which excludes the current chromosome in kinship calculations [71]. We standardized each phenotype and mapped with the Diversity Outbred outbreeding generation as an additive covariate and used the linear mixed-effects model with one kinship matrix per chromosome. We estimated the genome-wide significance thresholds by permuting the samples 1,000 times and performed a genome scan with each permutation. We retained the maximum $\log_{10}$ of the odds ratio (LOD) score from each permutation and estimated the genome-wide significance threshold of 7.6 from the 95th percentile of the empirical distribution of maximum LOD scores under permutation. We estimated the support interval around each peak using the 95% Bayesian Credible Interval.

For each peak with a LOD score above the genome-wide threshold > 7.6, we then searched for peaks associated with other traits that had LOD scores > 6 and confidence intervals that overlapped [72]. Our rationale was that the probability that a peak is biologically relevant, given that another trait has a co-located peak, is greater than the probability that a peak is significant with no prior evidence.

## Candidate gene selection

Within each QTL interval, we imputed the founder SNPs onto the Diversity Outbred mouse genomes using *qtl2* and performed association mapping. We selected the SNPs that were within a 1 LOD drop of the peak SNP in the QTL interval and filtered them to retain ones with missense, splice, or stop codon effects as annotated by the Sanger Mouse Genome Project [68]. We considered the genes in which these polymorphisms occurred as candidate causal genes for the associated trait(s).

## Trait-related gene sets

Because causal variants within a QTL may exert their influence through mechanisms other than gene expression, identifying differentially expressed genes within the QTL may be insufficient for ranking causal genes. Here we took an alternative approach and ranked candidates in

each QTL based on their predicted association with the mapped traits. To do this, we trained a Support Vector Machine (SVM) classifier to classify trait-related genes, and then used the trained SVM to score each positional candidate gene as trait-related or not-trait-related. We defined the training set of trait-related genes for the SVM as those genes that were highly correlated to the measured trait using the gene expression data described above. We calculated the Pearson correlation between the abundance of each transcript, and each physiological trait using rank Z normalized gene expression and traits. For each trait, we defined the training set of trait-related genes as the 500 genes with the largest magnitude Pearson correlation to the trait. We have made these gene lists available as a set of zipped text files in S1 File.

## Support Vector Machine classifier training

We trained SVMs to classify genes in each gene list as trait-related using features derived from the Functional Network of Tissues in Mouse [41]. The nodes in this network are genes, and the edges between them are weights between 0 and 1 that predict the likelihood that each pair of genes is annotated to the same Gene Ontology (GO) term or KEGG pathway [39]. Values closer to one indicate more certainty that the genes are more likely to be annotated to the same GO term or KEGG pathway and thus functionally related. The weights were derived using Bayesian integration of data sets from numerous sources of data, including gene expression, protein-protein interaction data, and phenotype annotations [41]. We used the top edges of the mouse lung network downloaded on March 31, 2021, from http://fntm.princeton.edu.

## Application of Support Vector Machine classifiers to identify genes functionally related to traits

We used SVMs to classify each positional candidate as trait-related or not-trait-related, as described previously [39,40]. Briefly, the expression-derived gene sets for each lung trait served as the *positive labeled set* of genes. We used the R package e1071 [73] to train SVMs to distinguish this set of genes from a balanced set of genes drawn randomly from the remaining genes in the lung network. The randomly selected genes were the *unlabeled set*. We performed this training 100 times, each time with a new set of random unlabeled genes. The SVMs were trained to distinguish positive labeled genes from unlabeled genes using the connection weights to the positive labeled genes. It is expected that positively labeled genes have strong connections to each other because they are functionally related. It is further expected that randomly drawn genes will be unrelated to the trait and to the positive labeled set and will thus have lower connection weights to the positive labeled genes. The SVM learns to distinguish these two groups of genes, and the resulting model can be used to classify genes that have not been seen before based on their connection weights to the positively labeled genes. We initialized each run by tuning the SVM over a series of cost parameters, starting with the sequence $10^{25}$ to $10^2$ by factors of ten, and iteratively narrowing the range until we found a series of eight cost parameters that maximized accuracy. In running each SVM, we used a linear kernel and 10-fold cross-validation.

We calculated the area under the receiver operating characteristic curves (AUC) for each set of trait-related genes as follows. We defined labeled positives (LP) as positive labeled genes that were classified by the SVM as trait related. Unlabeled negatives (UN) were unlabeled genes that were classified by the SVM as not trait related. Unlabeled positives (UP) were unlabeled genes that were classified as trait-related, and labeled negatives (LN) were positive labeled genes that were classified as not trait-related. These terms are conceptually like true/false positive and true/false negative scores. However, because unlabeled genes may not be truly unrelated to the trait, we cannot call them true negatives. Instead, we call them *unlabeled*.

We generated ROC curves using the *Unlabeled Predicted Positive Rate* (UPPR = UP/(UP +UN)), which is akin to the false positive rate, and the labeled positive rate (LPR = LP/(LP +UN)), which is akin to the false negative rate, along a series of SVM scores from the minimum to the maximum. We then calculated the average AUC across all 100 SVMs.

### Positional candidate scoring

After training SVMs for each trait, we scored all positional candidate genes in each QTL, defined as the minimum to the maximum position across a set of overlapping QTLs. Each candidate gene received one score for each trait that mapped to that location. To compare scores across traits, we used the UPPR for each gene at its calculated SVM score. The UPPR varies between 0 and 1, allowing us to compare scores for candidate genes across models. To visually compare across models, we used the -log10(UPPR) such that genes with small UPPR (high-nconfidence) got large positive values. In contrast, the SVM scores cannot be used to compare across models because they are unbounded and vary from model to model. Within each pleiotropic QTL, each gene received a score from each trait that mapped to the QTL.

### Mouse genome build and database versions

We used mouse genome build GRCm38 and SNPs and Indels from the Sanger Mouse Genomes Project, version 7, which uses Ensembl version 97 gene models. We also used and cross-referenced candidates with the Mouse Phenome Database GenomeMUSter, the Mouse Genome Informatics databases [74], and Ensembl's Variant Effect Predictor tool.

## Results

### Survival and body weight changes

We infected Diversity Outbred mice by aerosol with ~100 *M. tuberculosis* colony forming units in the first two experiments (N = 167), and ~25 colony forming units in the subsequent eight experiments (N = 683). Infection reduced survival of Diversity Outbred mice compared to identically housed, age-, gender-, and generation-matched uninfected Diversity Outbred mice and to identically housed age- and gender-matched infected C57BL/6J inbred mice (Fig 1A). Approximately one-third of infected Diversity Outbred mice succumbed prior to 60 days post infection (Fig 1A) reflecting early mortality between 20–56 days that peaked at 30 days (Fig 1B). This supersusceptible fraction of the Diversity Outbred population has been named Progressors [24,33,65,67]. Morbidity in all Progressors was due to pulmonary TB, confirmed by histology, recovery of viable *M. tuberculosis* bacilli from the lungs and absence of other diseases. After the first mortality wave subsided, cumulative survival declined slowly to nearly 600 days with no discernable mortality waves (Fig 1A and 1B). This resistant fraction of Diversity Outbred mice has been named Controllers [24,33,65,67]. The eight founder strains survived at least 40 days of *M. tuberculosis* infection, without early mortality (Fig 1A).

All mice were weighed prior to infection, during infection, and immediately before euthanasia. Non-infected Diversity Outbred mice gained weight until they developed other diseases or were euthanized at the experiment end (S1A Fig). Progressors gained weight for 2–3 weeks, and then lost weight rapidly (S1B Fig). Controllers and C57BL/6J inbred mice gained weight for long and variable durations through about 250 days of infection, and then most but not all slowly lost weight (S1C and S1D Fig). We questioned whether pre-infection body weight influenced differential susceptibility. Retrospective analysis identified no significant differences in pre-infection body weights of non-infected Diversity Outbred mice compared to Progressors; and a significant difference of on average 1.75 gm less in mean body weights of non-infected

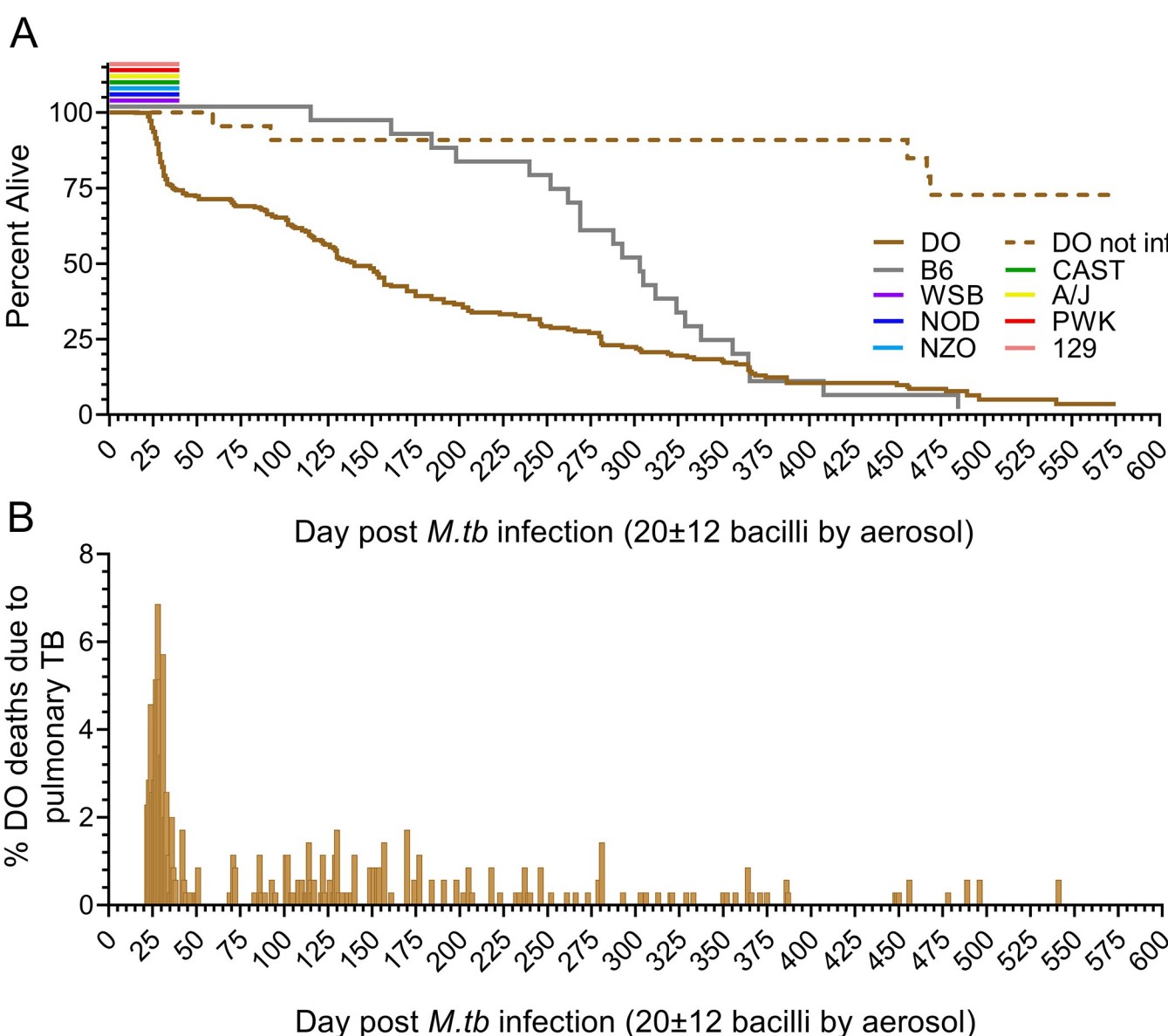

**Fig 1. Mouse survival following exposure to a low dose of aerosolized *M. tuberculosis* strain Erdman.** Diversity Outbred (DO) mice (n = 680, brown solid line), and the eight inbred founder strains (n = 15 to 78, colored lines) were infected with *M. tuberculosis* strain Erdman bacilli by aerosol. Panel A: Cumulative survival. Approximately 30% of the DO population succumbed to pulmonary TB by 60 days post infection and approximately 70% succumbed to between 60- and 600-days post infection. Of the eight inbred founder strains, survival studies were completed for the C57BL/6J inbred strain; the other seven inbred founder strains were euthanized 40 days after *M. tuberculosis* infection. No inbred founder or non-infected (NI) DO mice (n = 53, dashed line) showed mortality in that period. Panel B: Daily mortality of *M. tuberculosis*-infected DO mice, highlighting the early wave of mortality that peaked between 25- and 35-days post infection.

Diversity Outbred mice and Progressors compared to Controllers (S2A Fig). Whether this is spurious or biologically relevant (i.e., heavier pre-infection body weight is partially protective) remains to be determined. S2B, S2C, and S2D Fig show correlations between survival and eight clinical indicators. Seven indicators positively correlated with survival, including pre-infection body weight which was weakly positive. Only one indicator, the rate of body weight loss, negatively correlated with survival.

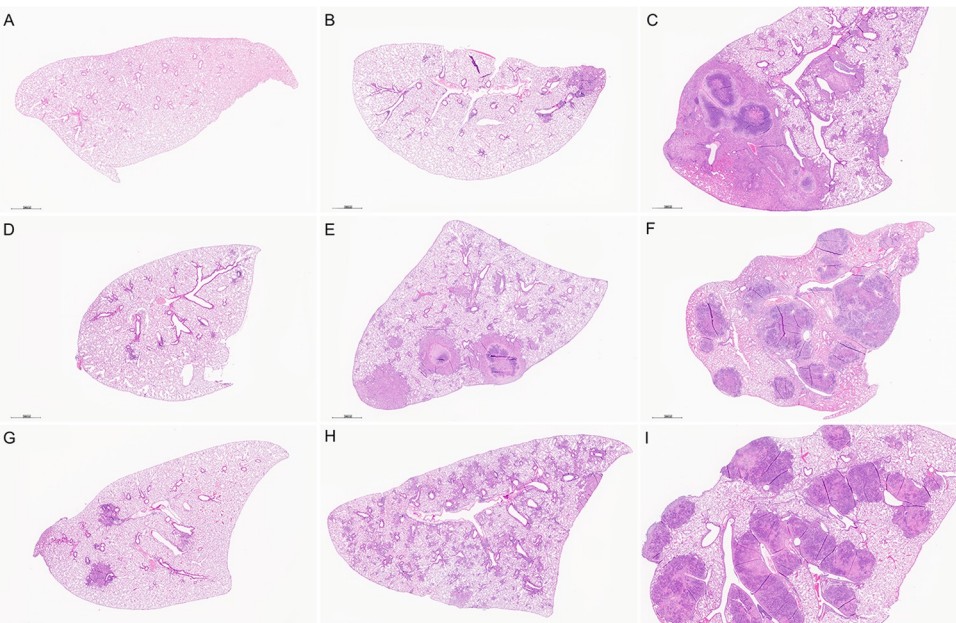

**Fig 2. Diversity Outbred (DO) mice develop a spectrum of histopathological lung lesions due to *M. tuberculosis* infection.** Lung lobes were formalin-fixed, paraffin-embedded, sectioned, and stained with hematoxylin & eosin. Panel A: Lung section from a non-infected DO mouse. Panels B through I: Lung sections from *M. tuberculosis*-infected DO mice euthanized eight weeks after infection show a spectrum of lesions from mild to severe (upper left to bottom right); focal lesions (e.g., Panel B) to diffuse infiltration (Panel H); and include necrotizing (Panels C, E, F, I) and non-necrotizing granulomas (Panels B, D, G, H). Low magnification (15X).

## Lung histology and automated image analysis of granuloma necrosis

By eight weeks of *M. tuberculosis* infection, Diversity Outbred mice showed a spectrum of lung lesions visible at low magnification (Fig 2) with variation in severity (minimal to marked); distribution of cellular infiltrates (focal, multifocal, and diffuse); and granuloma content (e.g., necrotizing, and non-necrotizing, shown in S3 Fig at higher magnification). Additional lesions included fibrin thrombosis with alveolar septal necrosis; cavities with peripheral fibrosis; foamy and multinucleated macrophages with cholesterol clefts; formation of secondary lymphoid follicles; alveolar septal fibrosis; and intra- and extracellular *M. tuberculosis* bacilli described elsewhere [19,24,28,67,75–77]. Since granuloma necrosis is a key feature of pulmonary TB in humans, we focused on this feature of disease and used our automated image analysis method [65] to quantify granuloma necrosis in lung tissue sections. The ratio of necrotic to non-necrotic lung tissue was then input as a trait for genetic mapping.

## Quantification of lung traits: M. tuberculosis burden and immune responses

We quantified *M. tuberculosis* lung burden by counting colonies from lung tissue homogenate and used the remaining lung homogenate to quantify cytokines and chemokines by ELISA [24,34]. Most lung traits were significantly greater in mice infected with *M. tuberculosis* compared to non-infected mice (Fig 3A), including neutrophil and monocyte/macrophage chemokines (CXCL1, CXCL2, CXCL5); mediators of innate immunity (S100A8, Tumor Necrosis Factor (TNF), interleukin (IL)-10, matrix metalloproteinase-8 (MMP8); mediators of acquired immunity (interferon-gamma (IFN-γ)) and *M. tuberculosis* burden. Pairwise Pearson correlation of all traits in infected mice showed that all the lung traits except IL10 and VEGF

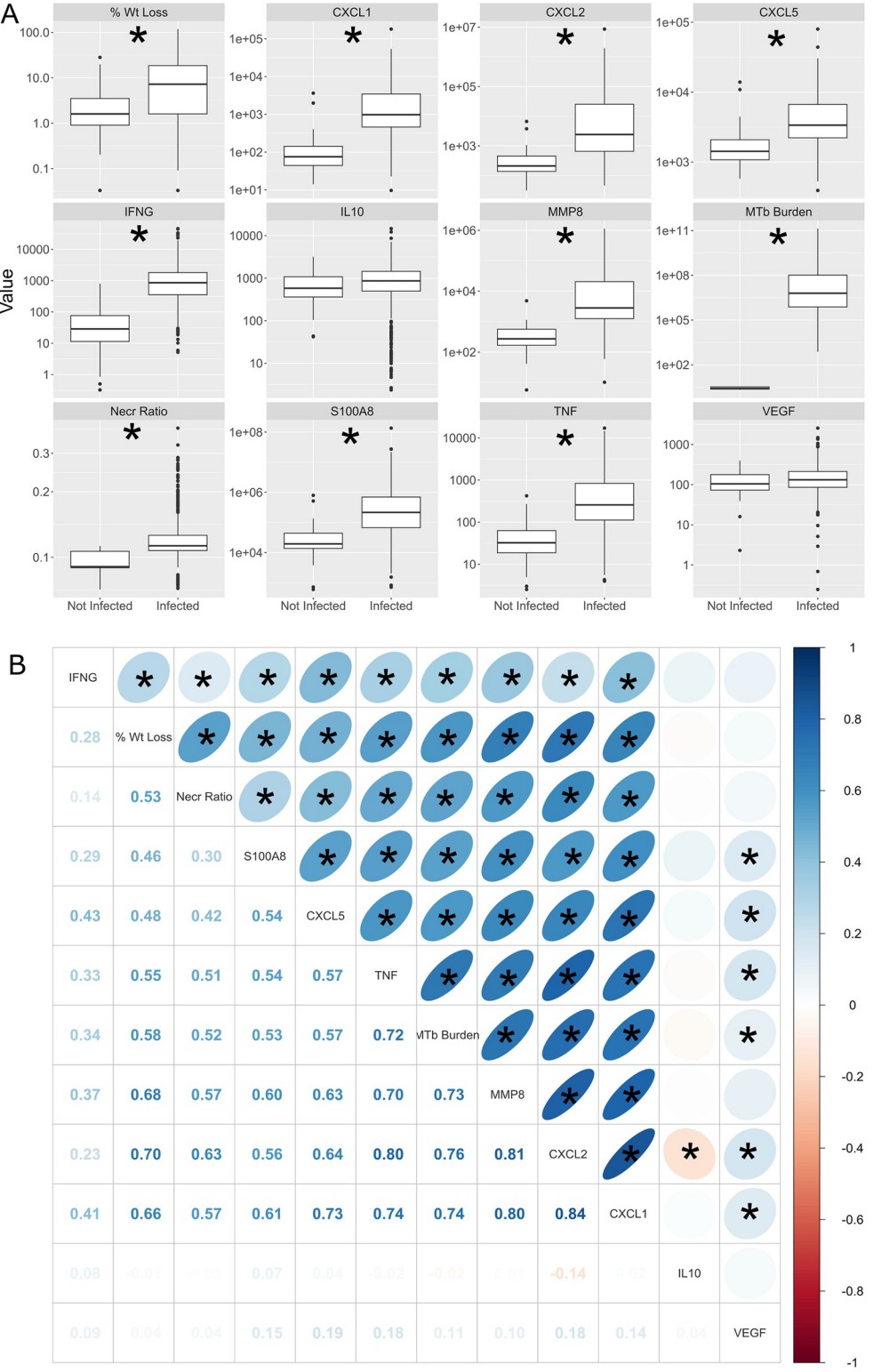

**Fig 3. *M. tuberculosis* infection of Diversity Outbred (DO) mice induces clinical, microbiological, and immune/inflammatory phenotypes that are positively and significantly correlated.** (A) Boxplots of phenotypes in non-infected (n = 77) and infected (n = 930) mice at euthanasia, which ranged from 5 to 553 days post-infection. Each box shows the median (central line) and the inter-quartile range in the box. The whiskers show 90% of the data and the dots beyond the whiskers show data points outside of the 90th percentile of the distribution. Significant differences at a

Bonferroni corrected p-value of 0.004 are shown with an asterisk. (B) Pearson correlation of selected phenotypes with n = 930 infected mice. Each cell in the upper triangle contains an ellipse in which the color and shape show the strength of the correlation. Positive correlations are shown in blue and negative correlations are shown in red. Strong correlations are shown by narrow ellipses, and weak correlations are shown by round ellipses. The more circular the ellipse, the closer the correlation is to zero. Numerical values of the Pearson correlation are shown in the lower triangle and range from -0.14 to 0.84. Phenotypes are ordered by hierarchical clustering of the similarity in correlations. Absolute Pearson correlations greater than 0.105 are significant at a Bonferroni-corrected p-value of 0.0007 and are shown with an asterisk.

positively correlated with each other (Fig 3B). Like previous findings in a small study of Diversity Outbred mice [24], correlations were strongest between *M. tuberculosis* lung burden and mediators of acute neutrophilic inflammation, innate immunity, and extracellular matrix degradation: CXCL1, CXCL2, TNF, and MMP8 with a mean correlation of 0.75.

## Overview of genetic mapping and gene prioritization within QTLs

We performed linkage mapping on each trait by regressing each on the additive founder allele dosage at each locus using the R package qtl2 [70]. We selected peaks with a permutation-derived significance threshold of 7.62 ($p_{GW} \leq 0.05$) and found eight peaks associated with multiple traits on chromosomes 1, 2, 3, 4, 16, and 17 (Table 1 and Fig 4). We observed that correlated traits colocalized to shared QTLs, and had similar patterns of allele effects, so we used a two-step procedure in which we recorded the confidence interval for peaks with a LOD $\geq 7.62$ and then looked for peaks of colocalized traits with a LOD $\geq 6.0$ ($p_{GW} \leq 0.6$). We reasoned that once we had found the first significant peak for one trait, the threshold for colocalized peaks with the same pattern of founder allele effects should be less, allowing refinement of the loci. Fig 5 shows a flow diagram of the types of input data for genetic mapping to identify QTLs, and the subsequent methods of gene prioritization.

**Table 1. Quantitative Trait Locus (QTL) mapping results of lung traits in *M. tuberculosis*-infected DO mice.** Six QTLs with LOD scores > 7.62 ($p_{GW} < 0.05$) and two with lower thresholds were identified by linkage mapping. Correlated traits with lower LOD thresholds ($\geq 6$) mapped to the same positions. The columns from left to right show: Lung trait(s), chromosome, LOD score of the peak, the position of the peak in megabases (Mb), lower and upper confidence intervals (C.I.) in Mb, and the width of the confidence interval in Mb. Five QTLs are new, and two QTLs are not new. *Dots3* overlaps with the QTL *tbs1*, previously identified by crossing A/Sn and I/St inbred mouse strains. *Dots7 and Dots8* overlap with *sst5* and *sst6*, previously identified by crossing C3HeB/FeJ and C57BL/6J inbred mouse strains.

| QTL | Trait(s) | Chr | LOD | Peak | C.I. Lo | C.I. Hi | C.I. Width | New |
|---|---|---|---|---|---|---|---|---|
| *Dots1* | *M.tb* burden<br>CXCL1<br>CXCL2<br>MMP8 | 1 | 7.68<br>7.87<br>$\geq 6$<br>$\geq 6$ | 155.36 | 154.25 | 156.71 | 2.45 | Yes |
| *Dots2* | CXCL2 | 2 | 7.69 | 118.65 | 118.05 | 118.86 | 0.82 | Yes |
| *Dots3* | S100A8 | 3 | 16.57 | 90.69 | 90.52 | 92.02 | 1.50 | No |
| *Dots4* | S100A8 | 4 | 7.64 | 22.43 | 22.18 | 23.79 | 1.61 | Yes |
| *Dots5* | *M.tb* burden<br>Wt loss<br>Granuloma necrosis | 16 | 8.45<br>$\geq 6$<br>$\geq 6$ | 38.28 | 33.28 | 43.28 | 10.00 | Yes |
| *Dots6* | *M.tb* burden<br>CXCL5 | 16 | 6.68 | 52.23 | 37.97 | 57.67 | 19.70 | Yes |
| *Dots7* | *M.tb* burden | 17 | ~6.5 | ~20 | ~12.5 | ~25 | ~12.5 | No |
| *Dots8* | Granuloma necrosis<br>Wt loss<br>CXCL1<br>MMP8<br>IL-10 | 17 | 8.12<br>$\geq 6$<br>$\geq 6$<br>$\geq 6$<br>$\geq 6$ | 35.02 | 33.94 | 41.06 | 7.12 | No |

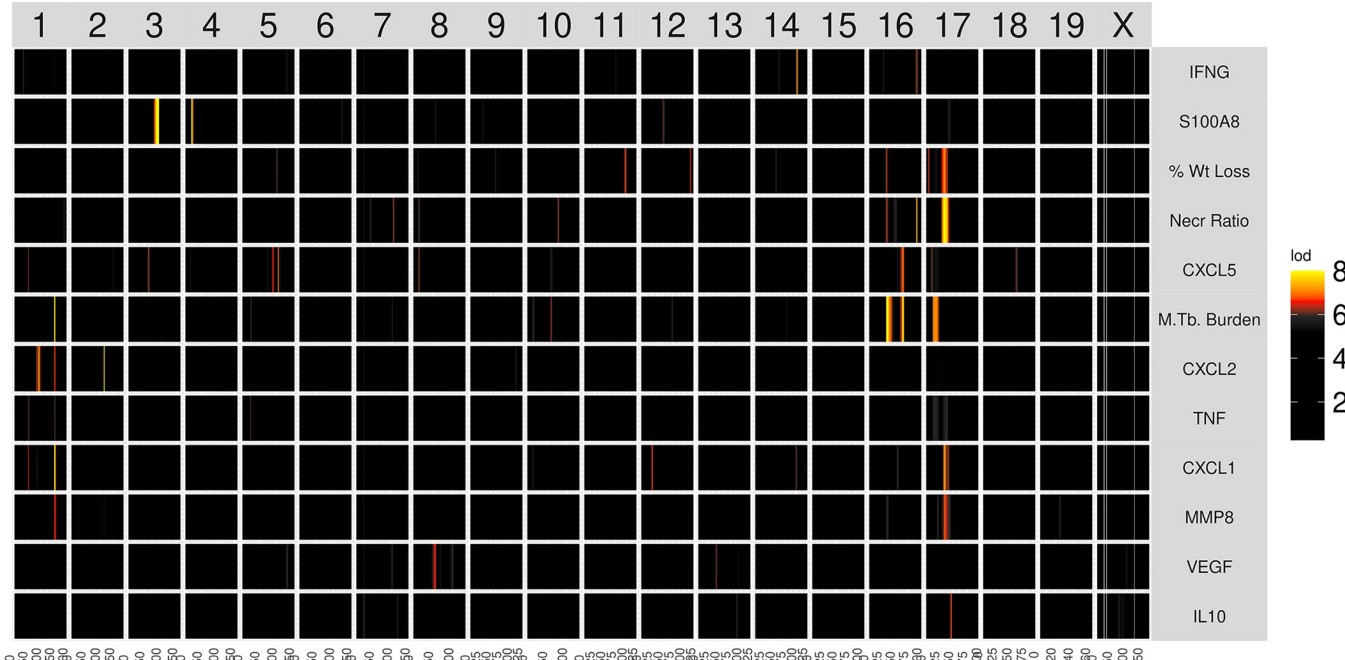

**Fig 4. The heatmap of linkage mapping peaks shows patterns of common genetic regulation.** Quantitative trait locus (QTL) mapping results from *M. tuberculosis* infected Diversity Outbred mice that were euthanized on or prior to 250 days post infection (if morbidity developed), and complete genotype and phenotypic trait data which also passed QA/QC (n = 853). The mouse genome, from chromosome 1 through X, is shown on the horizontal axis. Phenotypes are shown on the vertical axis. Each cell shows the logarithmic of the odds (LOD) score on one chromosome for the phenotype listed on the left, colored by the color scale. Higher LOD scores (oranges and yellows) meet or exceed statistical thresholds and show the significant genotype-phenotype associations. The phenotypes are hierarchically clustered based on the correlation between LOD curves, i.e., phenotypes with similar LOD curves are clustered next to each other.

### Chromosome 1: Diversity Outbred Tuberculosis Susceptibility locus 1 (Dots1)

*Dots1* is a new QTL on chromosome 1 with a peak LOD >7.6 ($p_{GW}$ < 0.05) at 155.36 M and interval 154.25–156.71 Mb shared by two correlated traits, *M. tuberculosis* burden and CXCL1 (Table 1). Two correlated traits (CXCL2 and MMP8) that mapped to the same position had LOD thresholds ≥ 6.0 ($p_{GW}$ ≤ 0.6) (Fig 4). Notably, these four correlated traits shared patterns of founder allele effects (S5 Fig), suggesting this QTL contains an important mechanism of genetic regulation for neutrophil-mediated activities, extracellular matrix remodeling, and *M. tuberculosis* growth. To refine the locus, we calculated the first principal component of those four traits and plotted the LOD curve, which also peaked between 154–156 Mb (Fig 6A) and plotted the founder allele effects. A/J, C57BL/6J and WSB/EiJ alleles contributed to greater values of principal component 1 and CAST/EiJ alleles contributed to lesser values (Fig 6B). We next imputed the founder SNPs onto the Diversity Outbred genomes and performed association mapping in a 10 Mb region around the peak (Fig 6C). Interestingly, the SNPs with highest LOD scores were outside of the peak, and none of the SNPs with the highest LOD scores were missense, stop, or splice site SNPs. This suggested the SNPs in the confidence interval could regulate expression of nearby genes, including some of the 47 protein-coding genes in the interval (Fig 6D and S1 File). To find and prioritize trait-related gene candidates within *Dots1*, we used the trained SVM model to rank gene candidates based on the strength of their functional relationship in gene expression network modules (S2 File). *Fam20b* and *Ncf2* ranked highest by functional scoring (Fig 6E). Table 2 shows known protein functions, MVAR genome annotations, allele effects, founder alleles containing SNPs, and predicted effects of missense SNPs in *Fam20b* and *Ncf2* genes on protein functions.

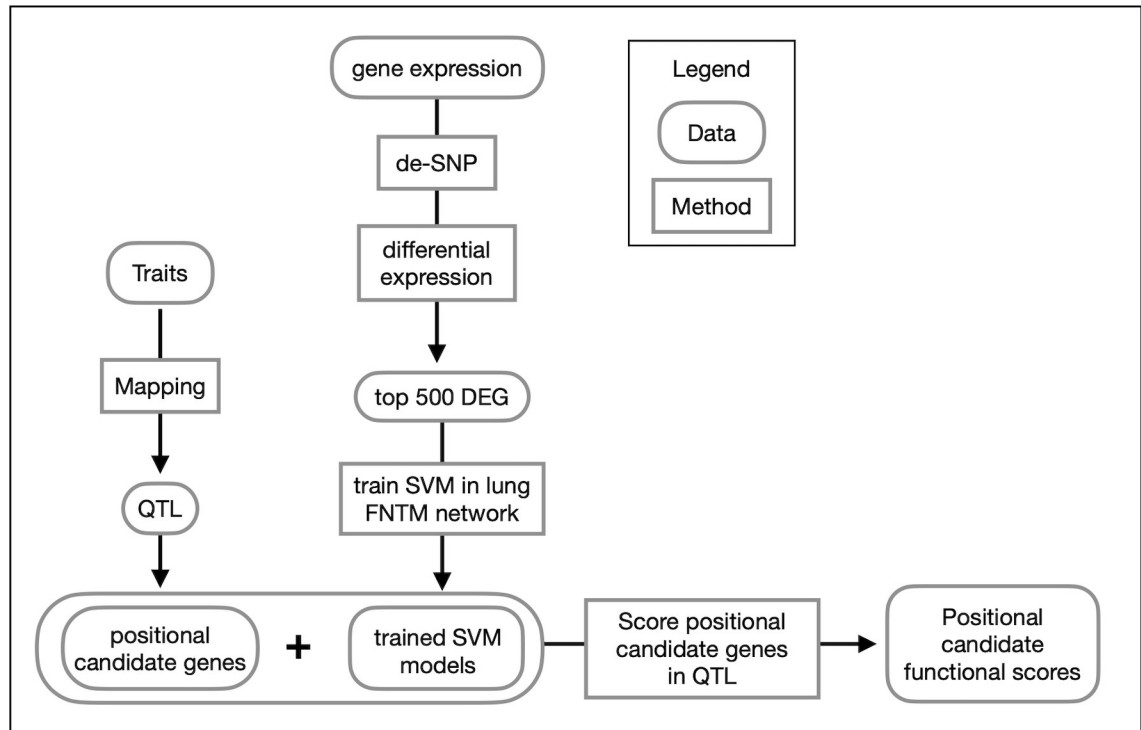

**Fig 5. Overview of gene prioritization methods.** Traits were mapped to identify positional candidate genes in QTLs. Gene expression data were analyzed for differential gene expression. The top 500 differentially expressed genes (DEG) were used to train SVMs to distinguish these trait-related genes from other genes in the genome using the FNTM mouse lung network. The fitted models were used to score positional candidates in each trait QTL. Positional candidates were then ranked as trait-related based on their functional scores.

## Chromosome 2: Diversity Outbred Tuberculosis Susceptibility locus 2 (*Dots2*)

*Dots2* is a new QTL, not shared by correlated traits (Table 1), and has a peak LOD of 7.69 ($p_{GW}$ < 0.05) at 22.43 Mb that was associated with lung CXCL2 protein levels (Fig 4). *Dots2* contains 19 protein coding genes (S1 File). Because this QTL was associated with only one trait, gene prioritization by functional scoring was not pursued.

## Chromosome 3: Diversity Outbred Tuberculosis Susceptibility locus 3 (*Dots3*)

*Dots3* is not a new QTL (Table 1) and overlaps with *tbs1*, a QTL previously identified by crossing A/Sn and I/St inbred mouse strains [78]. These strains are not founder strains of the Diversity Outbred population. *Dots3* was identified by a single peak with a high LOD of 16.57 at 90.69 Mb and interval 90.52–92.02 ($p_{GW}$ < $10^8$) associated with lung S100A8 (calgranulin A) protein levels (Figs 4 and 7A). CAST/EiJ alleles effects were high and PWK/PhJ allele effects were low (Fig 7B). SNPs with the highest LOD scores within the peak are shown (Fig 7C). The interval contains 12 protein coding genes (S1 File) including the *S100a8* gene (Fig 7D), suggesting that genetic variants which affect *S100a8* transcription regulate S100A8 (calgranulin A) protein levels in *M. tuberculosis* infection. Further, based on based on the strength of the functional relationship in gene expression network modules (S2 File) the trained SVMs identified *S100a8* as the gene with the highest functional score in *Dots3* (Fig 7E). *Dots3* also contains the

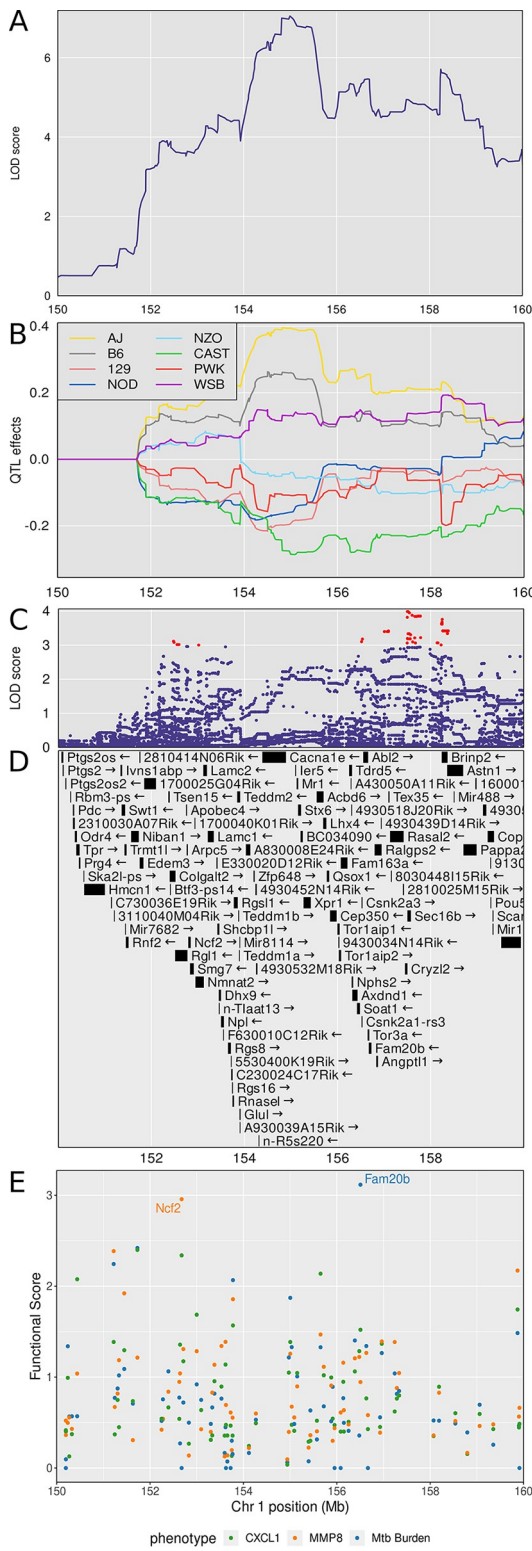

**Fig 6. Quantitative Trait Locus (QTL) mapping results of first principal component (PC1) of CXCL1, CXCL2, *M. tuberculosis* burden, and MMP8 identifies *Dots1* on chromosome 1, containing the gene candidates *Fam20b* and *Ncf2*.** Panel A: LOD curve for PC1 between 150 and 160 Mb on chromosome 1 with peak near 155.36 Mb. Panel B: Founder allele effects for PC1 in the same genomic interval. Each colored line is the best linear unbiased predictor for one of the founder alleles. Founder colors are shown in the upper left. Panel C: LOD score of the imputed SNPs in the

same genomic interval. Each point represents the LOD score of one imputed SNP. Panel D: Genes in the confidence interval. Panel E: Functional scores for genes in the chromosome 1 QTL. Each dot represents a single gene. Its position on the x axis is its position within the QTL. Its position on the y axis is the functional -log10(UPPR) derived from the SVM. Points are colored based on is correspondence with the trait—green with CXCL1, orange with MMP8, and blue with *M. tuberculosis* burden. *Fam20b* and *Ncf2* genes had the highest functional scores. QTL mapping results from n = 853 *M. tuberculosis* infected Diversity Outbred mice up to 250 days post infection.

gene *S100a9*, which encodes S100A9 (calgranulin B), a protein binding partner of S100A8 (calgranulin A) required to form the heterodimer, calprotectin. Table 2 shows known protein functions, MVAR genome annotations, allele effects, founder alleles containing SNPs, and predicted effects of missense SNPs in *S100a8* and *S100a9* genes on protein functions.

## Chromosome 4: Diversity Outbred Tuberculosis Susceptibility locus (*Dots4*)

*Dots4* is a new QTL, not shared by correlated traits (Table 1), and has a peak LOD of 7.64 ($p_{GW} < 0.05$) at 22.43 Mb and interval 22.18–23.79 Mb associated with lung S100A8 (calgranulin A) protein levels (Fig 4). *Dots4* contains two protein coding genes (S1 File). Because this QTL was associated with only one trait and contained few protein coding genes, prioritization by functional scoring was not pursued.

## Chromosome 16: Diversity Outbred Tuberculosis Susceptibility locus (*Dots5*)

*Dots5* is a new QTL on chromosome 16, shared by three correlated traits: lung *M. tuberculosis* burden (LOD = 8.45, $p_{GW} \leq 0.01$), weight loss, and granuloma necrosis with a peak at 38.3 Mb

**Table 2. Loci, candidate genes, protein functions, genome annotations, founder alleles with single nucleotide polymorphisms (SNPs), and predicted effects of SNPs on protein functions.** Predicted effects of missense SNPs on proteins were identified using Ensembl's Variant Effect Predictor and Sorting Intolerant from Tolerant (SIFT) databases. SIFT produces scores based on sequence homology and physicochemical similarity between amino acids encoded by SNPs. Normalized scores are based on the probability that the amino acid change is tolerated, with scores < 0.05 are predicted to be deleterious and all other scores are predicted 'tolerated'.

| QTL | Candidate | Known protein functions | MVAR genome annotations* | Alleles with SNPs** | Predicted effects of missense SNPs |
|---|---|---|---|---|---|
| *Dots1* | Ncf2 | superoxide production | Cn, Cs, I, U3, U5 | *129, **cast**, nod, nzo, pwk, wsb* | 2 of 2 tolerated |
| *Dots1* | Fam20b | glycosaminoglycan synthesis | Cn, Cs, I, U3, U5 | *129, **aj, cast**, nod, nzo, pwk, wsb* | No missense SNPs |
| *Dots3* | S100a8 | calgranulin A neutrophil responses | Cn, Cs, I, U3, U5 | *129, **cast**, nod, nzo, **pwk**, wsb* | 4 of 15 deleterious |
| *Dots3* | S100a9 | calgranulin B neutrophil responses | Cs, I | ***cast**, nod, nzo, **pwk**, wsb* | 1 of 1 tolerated |
| *Dots5* | Itgb5 | integrin subunit binds fibronectin | Cn, Cs, I, U3, U5 | *129, cast, **pwk**, wsb* | 2 of 6 deleterious |
| *Dots5* | Fstl1 | angiogenesis, wound-healing, fibrosis | Cn, Cs, I, U3, U5 | *aj, cast, **pwk**, wsb* | 3 of 4 deleterious |
| *Dots5* | Zbtb20 | transcriptional repressor | Not annotated | *aj, cast, nod, **pwk, wsb*** | Not available |
| *Dots8* | Ddr1 | interacts with collagen, fibrosis | Cn, Cs, I, U3, U5 | *129, aj, cast, nod, **nzo, pwk**, wsb* | No results retrieved |
| *Dots8* | Ier3 | signaling pathways, apoptosis | Cn, Cs, I, U3 | *aj, nod, **pwk**, wsb* | No results retrieved |
| *Dots8* | Vegfa | angiogenesis, endothelial activation | Cs, I, U3, U5 | *129, cast, nod, **nzo, pwk**, wsb* | No missense SNPs |
| *Dots8* | Zfp318 | transcriptional repressor in B cells | Cn, Cs, I, U3, U5 | *129, cast, nod, **pwk*** | 4 of 16 deleterious |

\* Mouse Variation Registry (MVAR) annotations

Cn Non-synonymous variant in codon, including missense.

Cs Synonymous variant in codon

I Variant in an intron

U3 UTR variant of the 3' UTR

U5 UTR variant of the 5' UTR

\*\* Bold alleles have high or low effects

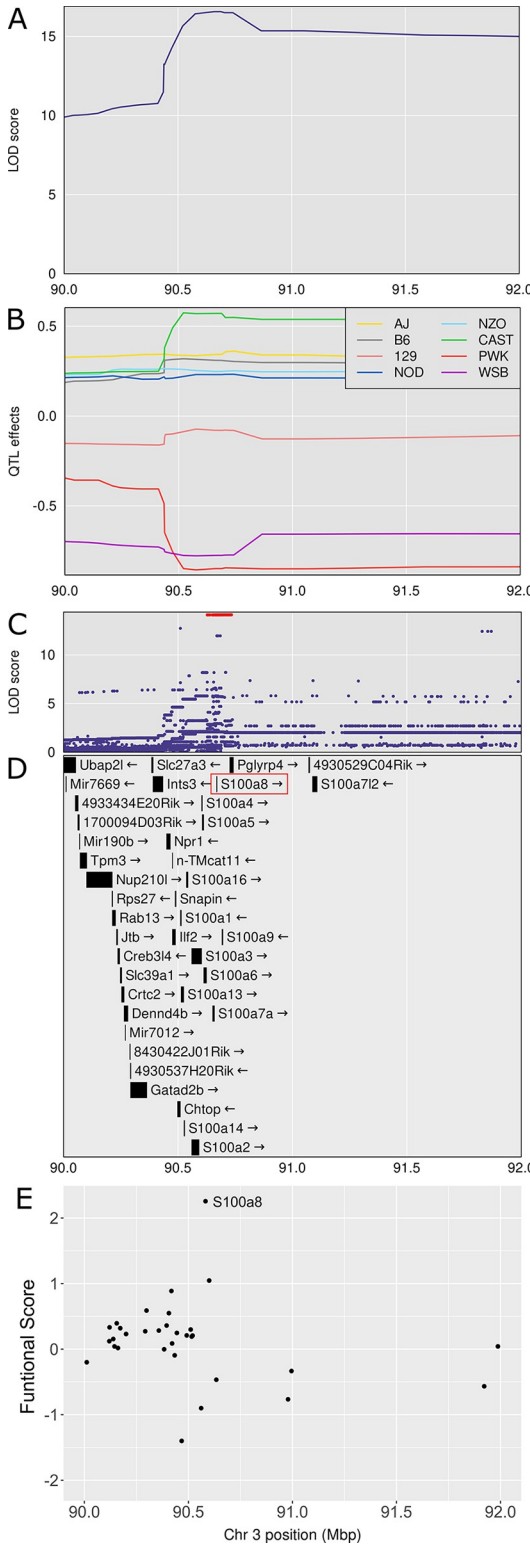

**Fig 7. Quantitative Trait Locus (QTL) mapping of lung S100A8 identifies *Dots3* on chromosome 3, containing the gene candidates *S100a8* and *S100a9*.** Panel A: LOD score in the confidence interval from 90 to 92 Mb on chromosome 3. Panel B: Founder allele effects within the confidence interval. Panel C: LOD score of the imputed SNPs with the highest SNPs colored in red. Panel D: Genes in the same interval. The gene *S100a8* is directly under the SNPs with the highest LOD scores. Panel E: Functional scores for genes in the chromosome 3 QTL. Each dot represents a

single gene. Its position on the x axis is its position within the QTL. Its position on the y axis is the functional -log10 (UPPR) derived from the SVM. The gene *S100a8* had the highest functional score. QTL mapping results from n = 853 *M. tuberculosis* infected Diversity Outbred mice up to 250 days post infection.

and interval 33.28–43.28 Mb (Table 1 and Fig 4). We calculated the first principal component of these traits and plotted the LOD curve showing its peak (Fig 8A). The founder allele effects indicate that C57BL/6J alleles contribute to greater values of principal component 1 and that PWK/PhJ and NZO/HILtJ alleles contribute to lesser (Fig 8B). We imputed the founder SNPs onto the Diversity Outbred genomes and performed association mapping around the peak, showing the SNPs with the highest LOD scores (Fig 8C). The SNPs with high LOD scores (Fig 8D) were not missense, stop, or splice site SNPs in the 75 protein coding genes within the interval (S1 File). By prioritizing genes based on functional relationships in network modules, we identified *Fstl1* and *Itgb5* as functional candidates associated with weight loss, and *Zbtb20* as a functional candidate associated with *M. tuberculosis* burden (Fig 8E and S2 File). Table 2 shows known protein functions, MVAR genome annotations, allele effects, founder alleles containing SNPs, and predicted effects of missense SNPs in *Fstl1*, *Itgb5*, and *Zbtb20* genes on protein functions.

## Chromosome 16: Diversity Outbred Tuberculosis Susceptibility locus (*Dots6*)

*Dots6* is a new QTL on chromosome 16 and shared by two correlated traits: *M. tuberculosis* burden and CXCL5 with a peak at 52.23 Mb and interval 37.97–57.67 Mb (Table 1 and Fig 4). The interval contains 101 protein coding genes (S1 File). Because the LOD score was less than LOD threshold 7.64 for significance ($p_{GW} < 0.05$), prioritization by functional scoring was not pursued.

## Chromosome 17: Diversity Outbred Tuberculosis Susceptibility locus (*Dots7*)

*Dots7* on chromosome 17 is not new, and overlaps with *sst5* and *sst6*, QTLs that were previously identified by crossing C3HeB/FeJ and C57BL/6J inbred mouse strains [79]. *Dots7* is associated with *M. tuberculosis* burden and the LOD peaks ~20 Mb (Table 1 and Fig 4). The interval contains 198 protein coding genes (S1 File). Because the LOD score was less than the LOD threshold 7.64 for significance ($p_{GW} < 0.05$), prioritization by functional scoring was not pursued.

## Chromosome 17: Diversity Outbred Tuberculosis Susceptibility locus (Dots8)

*Dots8* is not a new QTL and also overlaps with *sst5* and *sst6*, two QTLs that were previously identified by crossing C3HeB/FeJ and C57BL/6J inbred mouse strains [79]. Five traits: lung granuloma necrosis ("Necr Ratio"), weight loss, MMP8, CXCL1, and IL-10 mapped to *Dots8*. Lung granuloma necrosis had the highest LOD score (LOD = 8.12, $p_{GW} \leq 0.02$) at 35.02 Mb and interval 33.94–41.06 Mb Of those five traits, four positively correlated with each other (Fig 3) and had similar patterns of allele effects, and one, IL-10 had weak correlation and different pattern of founder allele effects.

We calculated the first principal component of the correlated traits and again performed QTL mapping. Principal component 1 mapped to a wide interval 30–50 Mb with a peak near 38 Mb (Fig 9A). The founder allele effects indicate PWK/PhJ alleles contribute to greater trait

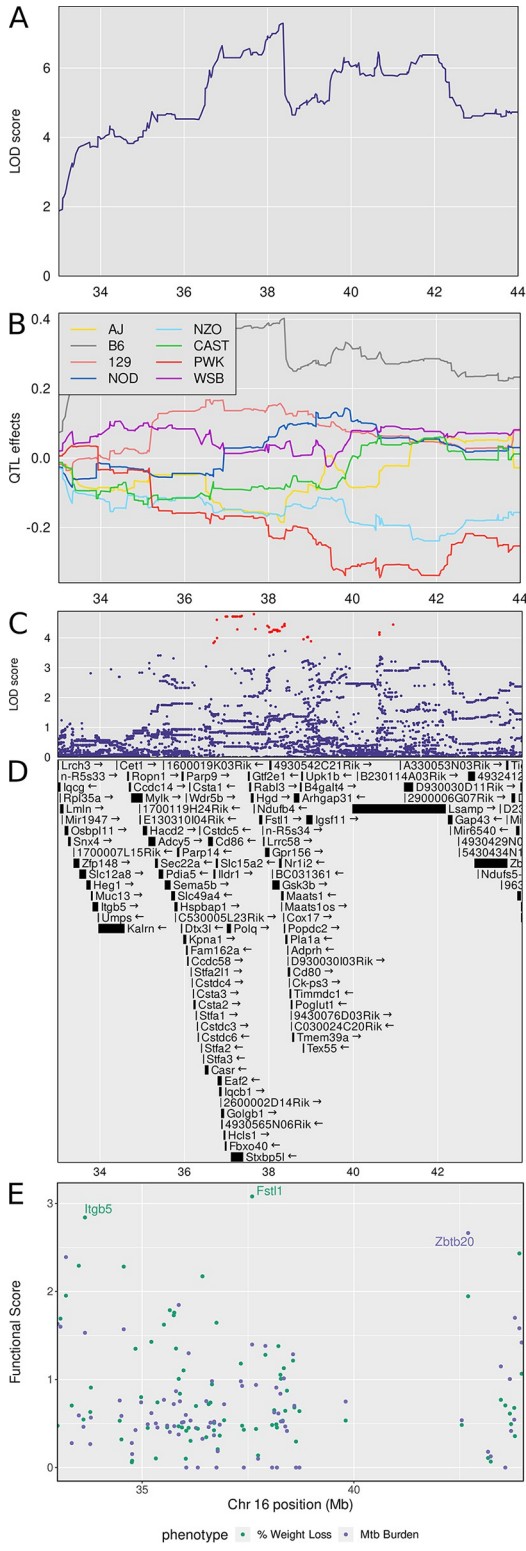

**Fig 8. Quantitative Trait Locus (QTL) mapping results of first principal component (PC1) of *M. tuberculosis* burden, weight loss, and granuloma necrosis identifies *Dots5* on chromosome 16, containing gene candidates *Fstl1*, *Itgb5*, and *Zbtb20*.** *M. tuberculosis* lung burden, weight loss, and granuloma necrosis map to a region on chromosome 16 near 38 Mb. Panel A: LOD profile for *M. tuberculosis* in the confidence interval. The genomic position on chromosome 16 is on the horizontal axis and the LOD score is on the vertical axis. Panel B: Founder allele effect in

the confidence interval. The vertical axis shows the estimates effect of gaining one founder allele. Panel C shows the SNP LOD score for association mapping using imputed SNPs. Panel D: Genes in the confidence interval. Panel E: Functional scores for genes in the chromosome 16 QTL. Each dot represents a single gene. Its position on the x axis is its position within the QTL. Its position on the y axis is the functional -log10(UPPR) derived from the SVM. Points are colored based corresponds to the trait—green with weight loss and blue with *M. tuberculosis* burden. *Fstl1* was the top ranked gene overall followed by *Itgb5* and *Zbtb20*. QTL mapping results from n = 853 *M. tuberculosis* infected Diversity Outbred mice up to 250 days post infection.

values, and NZO/HILtJ and NOD/ShiLtJ alleles contribute to lesser values (Fig 9B). We expected to find polymorphisms in the proximal peak of *Dots8* at 34–38 Mb because it contains the mouse histocompatibility-2 (H-2; or Major Histocompatibility Complex-II MHCII). This locus contains genes known to regulate innate and adaptive immunity and is known to be highly polymorphic. Indeed, the highest SNP association mapping LOD scores were over the mouse H-2 locus, located 36–38 Mb (Fig 9C), and there were 27 SNPs with protein-coding or splice site variation which occurred in 15 genes (S3 File). Among these were several histocompatibility genes (*H2-M1*, *H2-M5*, *H2-M9*, *H2-M11*) and several tripartite motif (TRIM) family genes (*Trim10*, *Trim26*, *Trim31*, *Trim40*).

The genes within the broad interval of *Dots8* are difficult to show in the figure and are provided in S1 File. There were 361 protein-coding genes within the 30–50 Mb locus. We prioritized positional candidate genes based on their functional relationships in network modules to the correlated traits (S2 File). This identified candidates *Ddr1*, *Ier3*, and *Vegfa* associated with CXCL1; and *Zfp318* associated with granuloma necrosis (Fig 9D). Table 2 shows known protein functions, MVAR genome annotations, allele effects, founder alleles containing SNPs, and predicted effects of missense SNPs in *Ddr1*, *Ier3*, *Vegfa*, and *Zfp318* genes on protein functions.

## Methodological and gene candidate validation

We performed three validation types shown in S6 Fig. This included (i) survival analysis of Diversity Outbred mice carrying PWK/PhJ alleles at the H-2 locus in *Dots8* on chromosome 17; (ii) quantification of S100A8 protein levels in lungs of *M. tuberculosis* infected PWK/PhJ and CAST/EiJ inbred founder strains; and (iii) infection of *S100a8* C57BL/6 gene deficient mice. Notably, infected Diversity Outbred mice carrying at least one copy of the PWK/PhJ allele at the mouse H-2 locus had shorter survival than mice carrying other alleles at the H-2 locus (S6A Fig).

The *Dots3* QTL on chromosome 3, associated with S100A8, had the highest LOD score, narrowest interval, distinct founder allele effects, and the gene candidate was the *S100a8* gene itself. For these reasons, the locus was selected for additional study. To help validate the method and ensure the founder allele effects generated by QTL mapping were biologically relevant, S100A8 was measured in lungs of the two founder inbred strains with the highest and lowest allele effects at the locus (Fig 7). As predicted by the allele effects, lungs from *M. tuberculosis* infected CAST/EiJ inbred mice contained significantly more S100A8 protein than lungs from *M. tuberculosis* infected PWK/PhJ inbred mice (S6B Fig). To evaluate *in vivo* effects of *S100a8* genetic deficiency, we obtained C57BL/6 breeding pairs to generate knockout, heterozygous, and wild-type mice. Littermates with null mutation (knockout), heterozygous, and wild-type C57BL/6 *S100a8* alleles were infected with *M. tuberculosis* for up to 50 days. The absence of one or both copies of the C57BL/6 *S100a8* allele had minimal impact on *M. tuberculosis* lung burden (S6C Fig). The lung granulomas of knockout, heterozygous, and wild type C57BL/6 mice were similar and non-necrotizing, composed of small dense aggregates of lymphocytes and macrophages, without neutrophils and lacking interalveolar pyknotic debris

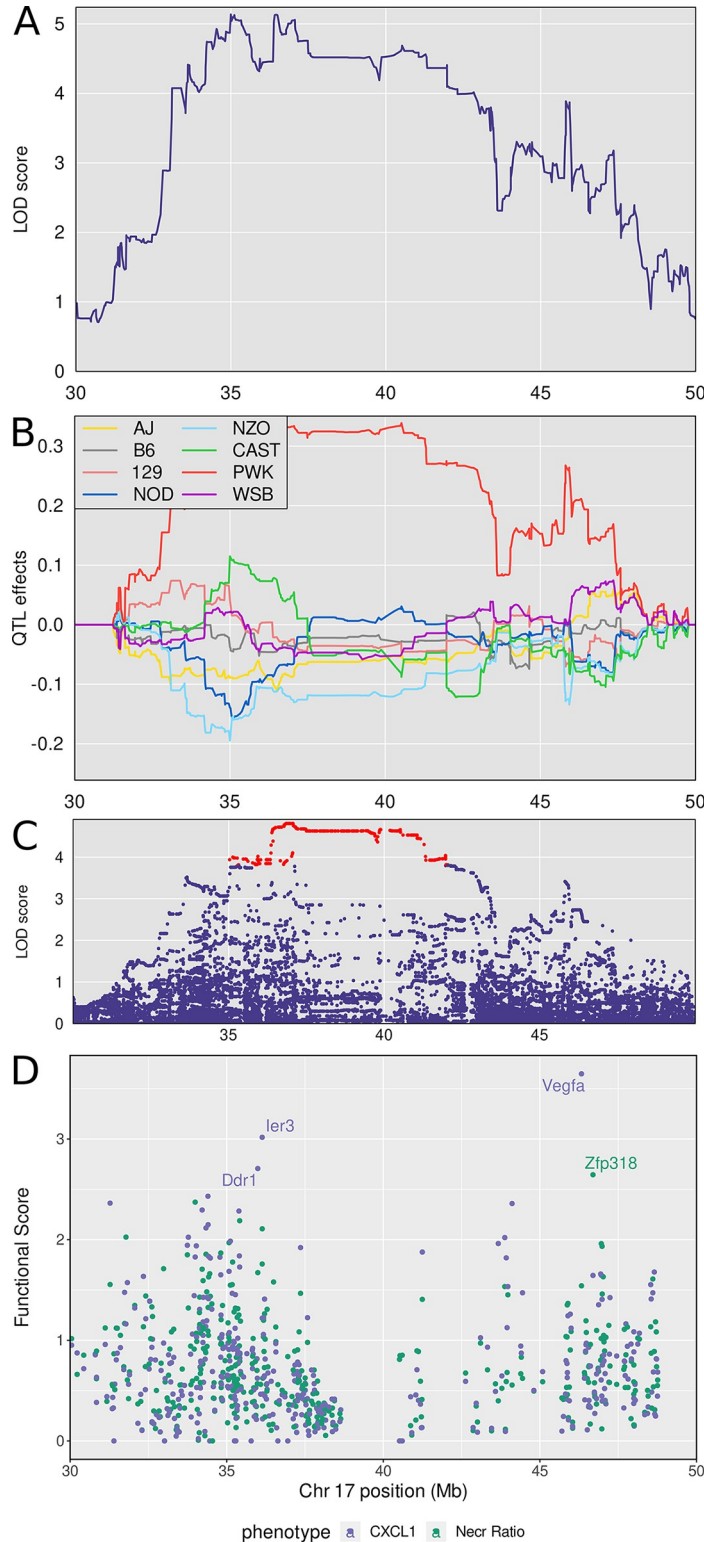

**Fig 9. Quantitative Trait Locus (QTL) mapping results of first principal component (PC1) of granuloma necrosis, *M. tuberculosis* burden, weight loss, CXCL1 and MMP8 identifies *Dots7* on chromosome 17, which contains gene candidates *Vegfa*, *Ier3*, *Ddr1*, and *Zpf318*.** Panel A: LOD score for PC1 of lung granuloma necrosis ratio, *M. tuberculosis* burden, MMP8, CXCL1, and weight loss in the interval where the phenotypes map. Panel B: Founder allele effects for the two peaks. Panel C: LOD score of the imputed SNPs in the interval, with the highest scoring SNPs

colored in red. Panel D: Functional scores for genes in the chromosome 17 QTL. Each dot represents a single gene. Its position on the x axis is its position within the QTL. Its position on the y axis is the functional -log10(UPPR) derived from the SVM. Points are colored based on correspondence to the traits—blue with CXCL1 and green with granuloma necrosis. *Vegfa* was the top ranked gene overall followed by *Ier3*, *Ddr1*, and *Zpf318*. QTL mapping results from n = 853 *M. tuberculosis* infected Diversity Outbred mice up to 250 days post infection.

(S6D, S6E and S6F Fig). This indicates that a lack or reduction of S100A8 has no major impact on early *M. tuberculosis* growth or granuloma formation in C57BL/6 inbred mice. These findings are unlikely to generalize to all genetic backgrounds, however, especially those genetic backgrounds predisposed to early accumulation of neutrophils and necrotizing granulomas with relatively high S100A8 levels, like the Diversity Outbred mice that are highly susceptible to *M. tuberculosis* [24,34].

## Discussion

TB remains a major public health concern in the United States and across the globe, with an estimated 2 billion people infected with *M. tuberculosis*; 8–9 million patients diagnosed each year, and 1–1.5 million deaths annually [2]. Fortunately, most humans (~90%) are highly resistant to *M. tuberculosis* and clear or control infection [11,80]. In susceptible adults, active pulmonary TB develops a few years following exposure and tends to occur in young to middle-aged adults in the prime years of their lives [81]. The disease is usually restricted to the lungs and is characterized by granuloma necrosis and cavitation, neutrophilic infiltration, and cachexia [82,83]. The genetic basis of pulmonary TB is complex and not attributable to single-gene defects that cause severe immune deficiency (i.e., Mendelian susceptibility to mycobacterial disease does not explain pulmonary TB) [15,84–89]. Although genome-wide association studies have identified loci, gene candidates, and SNPs associated with increased or decreased odds ratios for pulmonary TB, only a few (e.g., Ipr1/SP110b and HLA variants/I-A Major Histocompatibility genes) have been validated [16,90–96]. This has led investigators to seek alternative experimental mouse models such as Diversity Outbred mice and Collaborative Cross recombinant inbred strains to examine effects of genetics on host responses to *M. tuberculosis* [24–26,47].

An advantage of the Diversity Outbred mouse population is that infection with *M. tuberculosis* induces phenotypes that are rare in common laboratory inbred strains of mice [15–17]. Further, a growing body of evidence shows similarities in *M. tuberculosis*-infected Diversity Outbred mice and humans in biomarkers, gene expression signatures, and BCG vaccination [28,33,34,47,97]. The phenotypic similarities suggest that humans and Diversity Outbred mice may share underlying genetic pathways of immunity and disease. And, because SNP variants in the Diversity Outbred mouse genomes are dense, with balanced allele frequencies, any gene that plays a role in disease is theoretically detectable [98]. This eliminates a problem common to human genetic studies where under-represented alleles cannot be confidently associated with disease phenotypes because they are low-frequency genetic events.

We performed genetic mapping in *M. tuberculosis*-infected Diversity Outbred mice and used orthogonal methods to rationally select candidate genes. We first used DOQTL mapping, which relies entirely on phenotypic and genetic variation, to find eight QTLs on six different chromosomes named *Dots1* through *Dots8*. To refine loci, we then subjected the QTLs on chromosomes 1, 16, and 17 (*Dots1*, *Dots5*, *and Dots* 8) by mapping the first principal component of the correlated traits with similar patterns of allele effects that colocalized to the same interval. Finally, we applied a gene-based machine-learning SVM to identify and rank gene candidates based on functional scores. The sequential methods narrowed the candidate gene

list to eleven polymorphic, protein coding genes. Finally, the SNPs were critically examined using publicly available databases to find four candidates (*S100a8*, *Itgb5*, *Fstl1*, *Zfp318*) where missense SNPs are predicted to have deleterious effects on protein function.

All eleven candidates have roles in infection, inflammation, cell migration, extracellular matrix remodeling, or intracellular signaling. Of those, only one (mouse *Ncf2* in *Dots1* on chromosome 1) has a human homologue where a single SNP (G nucleotide in human NCF2 rs10911362) may provide a protective effect because this SNP was associated with a lower odds ratio for pulmonary TB in humans [99]. Absence of *Ncf2* in C57BL/6 inbred mice temporarily impairs resistance to *M. tuberculosis* infection by abrogating superoxide production, but the defect does not affect overall survival due to compensation by T cell mediated immunity [100]. Experimental validation of mouse *Ncf2* and human NCF2 polymorphisms remains to be confirmed.

We identified *Fam20b* in *Dots1* QTL on chromosome 1. The gene encodes a xylosylkinase that functions in glycosaminoglycan synthesis to produce extracellular matrix components in tissues. Deficiencies are embryologically lethal or cause cranioskeletal malformations [52,53,101,102]. A functionally homologous enzyme phosphorylates cadherins [103] which regulate immune cell migration [104] by interacting with extracellular matrix components, and since cell migration is required to form mycobacterial granulomas, *Fam20b* polymorphisms may alter host susceptibility to *M. tuberculosis* by changing extracellular matrix.

We identified *S100a8* and *S100a9* genes in *Dots3* QTL on chromosome 3, which encode S100A8 (calgranulin A) and S100A9 (calgranulin B). The proteins form monomers, homodimers, heterodimers, and multimers in inflammation, host defense, and nociception [105–108]. Some forms activate Toll-Like receptor 4; some activate the receptor for advanced glycation end-products (RAGE); and some sequester calcium, zinc, and manganese metal ions [105,109,110]. In pulmonary TB, S100A9 contributes to neutrophil localization to granulomas, and both S100A8 and S100A9 are biomarkers of TB-related lung damage [34,47,97,110–112]. Our results showing *S100a8* deficiency does not impact on *M. tuberculosis* lung burden, or granulomas, during early infection of C57BL/6 inbred mouse strain aligns with findings by Scott *et. al.* [97]. More specifically, the authors showed that S100A9 was elevated during chronic *M. tuberculosis* infection of C57BL/6 mice where it promotes neutrophil recruitment by CD11b, and C57BL/6 mice lacking S100A9 had lower *M. tuberculosis* CFU by 100 days post infection [97]. Whether these findings and mechanisms generalize to all genetic backgrounds and to all forms of S100A8 and S100A9 remain open questions, as S100A8 has anti-inflammatory functions [113] that appear independent from S100A9.

We identified *Itgb5* in *Dots5* QTL on chromosome 16 as a gene candidate. *Itgb5* encodes the beta 5 (β5) integrin subunit which dimerizes with the alpha v subunit to mediate cell adhesion and signaling by binding to fibronectin and vitronectin [49]. Notably, the β5 subunit is on the surface of cancer cells, and normal epithelial cells and activated endothelial cells but not on lymphoid or myeloid cells [49,114–120]. To our knowledge, neither the mouse nor human gene, nor subunit β5, nor the αvβ5 integrin heterodimer have been investigated in pulmonary TB.

We identified *Fstl1* in *Dots5* QTL on chromosome 16. The primary transcript encodes microRNA (miR)-198. The product is a secreted glycoprotein, FSTL1 with activities in angiogenesis, cell proliferation, differentiation, embryogenesis, metastasis, and wound healing; specifically reducing inflammation and fibrosis in cardiovascular disease [51,121–124]. Notably, *Fstl1* affects survival of *M. tuberculosis*-infected macrophages [43,44,125]. Given the central roles of macrophages, inflammatory mediators, and fibrosis in *M. tuberculosis* infection, understanding how *Fstl1* polymorphisms and FSTL1 function *in vivo* may inform TB pathogenesis, and host-directed therapy.

We identified *Zbtb20* in *Dots5* QTL on chromosome 16. The gene encodes a transcriptional repressor involved in glucose homeostasis; growth; hematopoiesis; innate immunity; neurogenesis; and B cell development and long-term survival of plasma cells [54–57,126–130]. Natural mutations occur in humans with Primrose Syndrome, although immune deficiencies are not reported [131]. To our knowledge, there are no studies on *Zbtb20* and *M. tuberculosis* infection or pulmonary TB. However, in *Listeria monocytogenes* infection, *Zbtb20*-deficiency improved CD8 T cell memory functions due to efficient use of diverse fuel sources [57]. Whether the same is true in pulmonary TB is unknown.

We identified *Ddr1* in *Dots8* QTL on chromosome 17. *Ddr1* encodes for the discoidin domain receptor 1 (DDR1), which interacts with collagen [50]. Initial studies suggested DDR1 function was restricted to epithelial cells; however, recent work shows expression on solid tumors, metastatic cells, and mouse histiocytic cancer cell lines, J774 and Raw264.7 [132–138]. DDR1 has additional roles in demyelination, fibrosis, vitiligo, and wound healing, and it is also a promising target for anti-fibrotic therapy [139–143]. Whether *Ddr1* (mouse) or DDR1 (human) gene polymorphisms contribute to pulmonary TB, or whether it could be a target for anti-fibrotic therapy in TB are areas open for investigation.

We identified the immediate early response gene, *Ier3* in *Dots8* QTL on chromosome 17. *Ier3* transcription is triggered by cytokines, hormones, DNA damage, and infections. The protein, IER3, regulates apoptosis, DNA repair, differentiation, and proliferation by interfering with NF-κB, MAPK/ERK and PI3K/Akt signaling pathways [58–60,144–146]. Mice lacking *Ier3* are more susceptible to *Leishmania* [147], an intracellular pathogen that shares some similar immune responses profiles with those induced by *M. tuberculosis* but we did not find studies showing that mutated *Ier3* also increases susceptibility to *M. tuberculosis*. One *in vitro* study of human macrophages, however, had high levels of IER3 mRNA following infection with a hypervirulent strain of *M. tuberculosis* [148] indicating the transcriptional pathway is triggered.

We identified *Vegfa* in *Dots8* QTL on chromosome 17. Mouse *Vegfa* and human VEGFA, encode for a heparin-binding protein and essential growth factor that induces proliferation, migration, and permeability changes in vascular endothelial cells by binding VEGFR1 and VEGFR2 [149–152]. Roles for VEGF in pathogenesis and diagnostics for extrapulmonary TB, cavitary TB, and active TB have been published [153–155]. Myeloid-specific gene deletion of *Vegfa* extended survival of C57BL/6J inbred mice infected with *M. tuberculosis* [156], highly noteworthy because very few gene deletions improve survival. Whether *Vegfa* or VEGFA polymorphisms have the same effect is unknown.

Lastly, we identified *Zfp318* in *Dots8* QTL on chromosome 17. The gene encodes the transcription factor, zinc finger protein 318 and it is expressed in testes, hematopoietic, and lymph nodes [157]. In B cells, the protein represses transcription required for class switching, helping to maintain B cell anergy and prevent autoimmunity [158–160]. Database and literature searches identified no publications on mouse *Zpf318* or human ZPF318 in infectious diseases.

When we compared genetic mapping results from Diversity Outbred mice to results from colleagues using Collaborative Cross inbred strains [26], QTLs and gene candidates did not overlap although we measured a few of the same traits by standard laboratory methods (e.g., body weight changes, lung *M. tuberculosis* burden, and lung CXCL1 by immunoassays). This suggests that phenotype-genotype relationships in the Collaborative Cross strains may be fundamentally different than in the Diversity Outbred mouse population, possibly due to heterozygosity in the Diversity Outbred population which is absent from Collaborative Cross inbred strains. Other reasons could be differences in routes of infection that change the host cell types first encountering *M. tuberculosis* bacilli which alters antigen presentation, T-cell, and B-cell priming. Here, we modeled natural aerosol exposure by delivering a low dose of *M.*

*tuberculosis* bacilli to the lungs of Diversity Outbred mice in nebulizer-delivered aerosol mist, and then focused on quantification of lung disease. In contrast, Smith *et al* [26] took a different approach by using intravenous infection with $1\times10^6$ bacilli to take advantage of their rich library of transposon mutants, allowing detailed assessment of pathogen-associated QTLs. As the intravenous route of infection favors rapid induction of acquired immunity by delivering bacilli directly to lymphoid organs (i.e., spleen, thoracic, and abdominal lymph nodes by portal and systemic circulation), this approach maximized identification of unique Host-Interacting-with Pathogen QTLs and resulted in a prioritized list of candidate genes involved in immunity.

Overall, by using a systems genetics approach focused on the lungs, we identified multiple new and existing QTLs, and 11 candidate genes. Of those, gene products for five (*Ncf2*, *Fstl1*, *Zbtb20 Vegfa*, *Zfp318*) have known roles in recruitment, activation, or regulation of effector functions of immune cells (e.g., neutrophils, monocytes, macrophages and CD8 T cells). The gene products for three candidates (*Fam20b*, *Itgb5*, *Ddr1*) have known roles in epithelial cell, endothelial cell, and (possibly) macrophage adhesion to extracellular matrix glycoproteins or participate in remodeling of extracellular matrix. The gene products for two candidates (*S100a8* and *S100a9*) have complex and context-dependent roles in innate immune response signaling and in host defenses. Finally, the gene product of one candidate (*Ier3*) controls early stress responses of cells, including cell survival and death pathways. Ten of the eleven candidates have annotated polymorphisms; six have missense SNPs in protein coding regions; and the SNPs in four candidates (*S100a8*, *Itgb5*, *Fstl1, and Zfp318)* are predicted to have deleterious consequences on protein functions. Together, these results yield a concise list of candidates that may be major regulators of host necrotizing and inflammatory responses during *M. tuberculosis* infection and pulmonary TB disease progression. Future studies will focus on testing effects of these gene candidates and polymorphisms *in vivo* and identification of pathogenic molecular and cellular mechanisms.

## Supporting information

**S1 Fig. Mouse body weight following a low dose of aerosolized *M. tuberculosis* strain Erdman.** Mice were infected with a low dose of *M. tuberculosis* strain Erdman by aerosol and infection progressed unmanipulated until mice were euthanized due to IACUC-approved morbidity criteria. Panel A: Body weight changes of identically housed, age-, gender-, and generation-matched non-infected Diversity Outbred (DO) mice (n = 49) compared to baseline. Panels B, C, and D: Body weight changes of Progressor DO mice (n = 195); Controller DO mice (n = 145); and C57BL/6J inbred founder strain mice that succumbed to pulmonary TB (n = 39), are shown over time compared to pre-infection baseline. All mice were weighed 1 to 3 days prior to *M. tuberculosis* infection, at least twice per week during infection, and immediately before euthanasia. Each line is the body weight expressed as a percent change compared to initial pre-infection body weight.
(TIF)

**S2 Fig. Clinical correlates of survival due to pulmonary TB in Diversity Outbred (DO) mice following exposure to a low dose of aerosolized *M. tuberculosis* strain Erdman.** Age-, gender-, and generation-matched DO mice were assigned to cages at random, and infected (or not infected) with a low dose of *M. tuberculosis* strain Erdman by aerosol exposure. All mice were initially weighed 1–3 days prior to infection, at least twice per week during infection, and immediately before euthanasia. Panel A: shows retrospective analysis of pre-infection body weights of Non-infected mice (n = 76) compared to pre-infection body weights of Progressors (n = 298) and pre-infection body weights of Controllers (n = 195), shown as box-and-whisker plots with the line at the mean for each group, and whiskers at the minimum and maximum.

Data were analyzed by 1-way ANOVA with Tukey's multiple comparisons test ***p<0.001; ****p<0.0001. Panels B, C, D: Infection progressed unmanipulated until mice were euthanized due to IACUC-approved morbidity criteria. Panel B: The rate of weight loss (gm/day) and duration of body weight (BW) loss in days negatively correlate with each other. Panel C: Duration of BW loss was strongly, positively, and linearly correlated with survival by Spearman correlation analysis (r = 0.848 with dashed lines indicating the 95% confidence interval, 0.8204 to 0.8717, p<0.0001). Panel D: Correlation matrix to show how survival and eight clinical indicators correlate with each other. Only correlations with p-values <0.00001 are shown on the matrix. Cells marked by an "X" were not significantly correlated.
(TIF)

**S3 Fig. Examples of necrotizing and non-necrotizing lesions in lungs of *M. tuberculosis* infected Diversity Outbred (DO) mice by light microscopy at high magnification.** Lung lobes were formalin-fixed, paraffin-embedded, sectioned, and stained with carbol fuschin and counterstained with hematoxylin & eosin. Panels A and B: High magnification images of necrotizing lung lesions. One example contains abundant pyknotic nuclear debris (A) and one example contains abundant fibrin, eosinophilic cellular debris, and less nuclear debris (B). Panels C and D: High magnification images of non-necrotizing lung lesions. Both examples contain mostly viable cells, including macrophages, foamy macrophages, and foci of lymphocytes (400X).
(TIF)

**S4 Fig. Similar patterns of founder allele effects for four traits that map to distal chromosome 1 QTL.** The pattern of founder allele effects of four correlated traits all with LOD > 6.0 on chromosome 1 at 155.36 Mb is similar. Panel A: Founder allele effects for CXCL1. Panel B: Founder allele effects for CXCL2. Panel C: Founder allele effects for MMP8. Panel D: Founder allele effects for *M. tuberculosis* burden. Founder strain names are on the horizontal axis and the standardized allele effect are on the vertical axis. Allele effects from QTL mapping results from n = 853 *M. tuberculosis* infected Diversity Outbred mice up to 250 days post infection.
(TIF)

**S5 Fig. Receiver operator characteristic (ROC) curves for SVM training on traits used in gene prioritization.** Each panel shows the true positive rate of the trained SVM as a function of the false positive rate for each trait. The area under the curve (AUC) is noted for each panel.
(TIF)

**S6 Fig. Validation of QTL mapping results.** Panel A: *M. tuberculosis* infected Diversity Outbred (DO) mice with one or more PWK/PhJ alleles (red) at the H-2 locus on chromosome 17 in *Dots8* (near 36 Mb) have significantly reduced survival compared to DO mice carrying other alleles (grey) (p = 0.00075, Cox-PH test). Days of survival are shown on the horizontal axis and the proportion of mice surviving is shown on the vertical axis. Panel B: Lungs from *M. tuberculosis* infected CAST/EiJ inbred mice contain significantly more S100A8 protein (calgranulin A) than lungs of PWK/PhJ inbred mice at the time points indicated on the horizontal axis. PWK/PhJ (red) and CAST/EiJ (green) inbred founder strains with 4–6 mice per strain per time point, analyzed by Mann-Whitney *t*-tests within each time point, *p<0.05. Panel C: *M. tuberculosis* infected *S100a8* knockout (KO), heterozygotes (HET), wild-type (WT) C57BL/6 inbred mice were euthanized at the time points indicated, and *M. tuberculosis* lung burden assessed by CFUs with total combined 15–22 mice per genotype per time point from 2 independent experiments, shown as average and standard error of the mean. No significant (ns) differences were identified within each time point by mixed effects ANOVA with Tukey's post-test (p<0.05). Panel D, E, F: Representative examples of lymphohistiocytic non-

necrotizing granulomas day 50 post infection from *S100a8* knockout (D), heterozygous (E), and wild type (F) C57BL/6 inbred mice magnified 400x normal.
(TIF)

**S1 File. This file is an Excel workbook containing worksheets that list all protein-coding genes in each QTL with the functional candidates highlighted.**
(XLSX)

**S2 File. This is an Excel file that lists the top ten functional candidates for each trait in each QTL.**
(XLSX)

**S3 File. This is an Excel file that lists genes within chromosome 17 QTL.**
(XLSX)

## Acknowledgments

We thank Ms. Julie Tzipori, Mr. Curtis Rich, Mr. Donald Girouard, and Dr. Sam Telford III for study support at the New England Regional Biosafety Laboratory at Tufts University Cummings School of Veterinary Medicine, North Grafton, MA. Ms. Frances Brown, Ms. Linda Wrijil, Ms. Sarah Ducat, Ms. Gina Scarglia, Ms. Dian Taylor provided histology services at Tufts University's Cummings School of Veterinary Medicine Comparative Pathology and Genomics Shared Resource, directed by Dr. Amanda Martinot and Dr. Heather Gardner. The Digital Histology Shared Resource at Vanderbilt University Medical Center performed whole slide imaging. The following reagents were obtained through BEI Resources: ESAT-6, Recombinant Protein Reference Standard, NR-49424; CFP-10, Recombinant Protein Reference Standard, NR49425; Plasmid pMRLB.7 Containing Gene Rv3875 (Protein ESAT-6) from Mycobacterium tuberculosis, NR-50170; Plasmid pMRLB.46 Containing Gene Rv3874 (Protein Cfp10) from *Mycobacterium tuberculosis*, NR-13297; *Mycobacterium tuberculosis*, Strain H37Rv, Culture Filtrate Proteins, NR-14825; and *Mycobacterium tuberculosis*, Strain H37Rv, Cell Wall Fraction, NR-14828. All microarray protocols were conducted by the Boston University Microarray and Sequencing Resource (BUMSR), and we thank Mr. Eduard Drizik of the BUMSR for initial microarray analyses. The following individuals are thanked for their technical expertise: Ms. Victoria Mello at Tufts Cummings School of Veterinary Medicine, North Grafton, MA; Mr. Austin Hossfeld at The Ohio State University, Columbus, OH; and Dr. Joanne Turner at Texas Biomedical Research Institute, San Antonio, TX.

Boston University Microarray and Sequencing Resource received support from the National Institutes of Health UL1TR001430. The New England Regional Biosafety Laboratory at Tufts University's Cummings School of Veterinary Medicine received support from the National Institutes of Health UC6A1066843. Tufts University's Cummings School of Veterinary Medicine Comparative Pathology and Genomics Shared Resource received support from the Massachusetts Life Sciences Center.

## Author Contributions

**Conceptualization:** Daniel M. Gatti, Sherry L. Kurtz, Karen L. Elkins, Igor Kramnik, Gillian Beamer.

**Data curation:** Daniel M. Gatti, Deniz Koyuncu, Thomas Tavolara, Adam Gower, Gillian Beamer.

**Formal analysis:** Daniel M. Gatti, Anna L. Tyler, J Matthew Mahoney, Adam Gower, Gillian Beamer.

**Funding acquisition:** Daniel M. Gatti, Bulent Yener, Metin N. Gurcan, Gillian Beamer.

**Investigation:** Daniel M. Gatti, Anna L. Tyler, Melanie L. Ginese, Anas Alsharaydeh, Gillian Beamer.

**Methodology:** Daniel M. Gatti, Anna L. Tyler, J Matthew Mahoney, Gary A. Churchill, Denise Dayao, Emily McGlone, Melanie L. Ginese, Aubrey Specht, Philipe A. Tessier, Gillian Beamer.

**Project administration:** Gillian Beamer.

**Resources:** Gary A. Churchill, Philipe A. Tessier, Gillian Beamer.

**Supervision:** MK Khalid Niazi, Gillian Beamer.

**Visualization:** Daniel M. Gatti, Gillian Beamer.

**Writing – original draft:** Daniel M. Gatti, Anna L. Tyler, J Matthew Mahoney, Deniz Koyuncu, Adam Gower, Anas Alsharaydeh, Gillian Beamer.

**Writing – review & editing:** Daniel M. Gatti, Anna L. Tyler, J Matthew Mahoney, Bulent Yener, Deniz Koyuncu, Metin N. Gurcan, MK Khalid Niazi, Thomas Tavolara, Adam Gower, Sherry L. Kurtz, Gillian Beamer.

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
