## [Decision Letter · Decision Letter 0]

22 Feb 2024

Dear Dr. Beamer,

Thank you very much for submitting your manuscript "Systems genetics uncover new loci containing functional gene candidates in Mycobacterium tuberculosis-infected Diversity Outbred mice." for consideration at PLOS Pathogens. As with all papers reviewed by the journal, your manuscript was reviewed by members of the editorial board and by several independent reviewers. In light of the reviews (below this email), we would like to invite the resubmission of a significantly-revised version that takes into account the reviewers' comments.  No new experiments are required but significant changes to editorial and data presentation is needed in the revised submission.  

We cannot make any decision about publication until we have seen the revised manuscript and your response to the reviewers' comments. Your revised manuscript is also likely to be sent to reviewers for further evaluation.

Sincerely,

Padmini Salgame

Academic Editor

PLOS Pathogens

Michael Wessels

Section Editor

PLOS Pathogens

Michael Malim

Editor-in-Chief

PLOS Pathogens

orcid.org/0000-0002-7699-2064

Reviewer's Responses to Questions

**Part I - Summary**

Reviewer #1: Apparently. this is the most comrehensive whole genome mapping of QTLs involved in TB infection control in mice. The authors used diverse outbred (DO) mouse colony which provides a better approximation to genetic diversity observed in natural populations; thus, the results obtained are specifically valuable. They report about several newly mapped TB-related QTLs, and confirm genomic locations of a few QTLs mapped in earlier studies. Overall, the manuscript adds an amazing amount of novel information on quantitative host TB genetics.

Reviewer #2: Gatti et al report the findings of a large systems genetics study with Mtb-infected Diversity Outbred mice. A total of 850 Diversity Outbred mice were Mtb-infected. In addition to haplotyping each mouse against 137,302 SNP/genetic markers, the mice were scored for survival time, weight loss, lung histopathology (with necrosis specifically scored), Mtb lung CFU counts, and lung levels of 11 different cytokines/chemokines.

Even though the 8 founder strains do not display early mortality, interestingly one-third of the Diversity Outbred mice died within 60 days with necrotic granulomas. Thereafter there was no pattern to the survival histogram with some mice surviving to approximately 600 days.

The authors identified eight Diversity Outbred TB susceptibility loci (Dots1-Dots8), five of which have not previously been associated with TB disease progression. Validation was performed in three ways: (i) on mice with Dots8 (survival), (ii) by measurement of S100A8 (associated with Dots3 and Dots4) levels in two founder lines, and (iii) evaluating the survival of s100a8 genetic knockout mice (+/+, +/-, and -/-) mice after Mtb infection.

The authors have responded to a previous round of reviews (3 reviewers) with modifications that are helpful and thorough, and these previous revisions improve the manuscript. I only have a few minor items for the authors to consider.

Reviewer #3: Gatti et al utilize the Diversity Outbred mouse model of TB and a machine learning strategy to identify new loci and candidate genes that correlate to deleterious immune outcomes. By focusing on drivers of granuloma necrosis and inflammation, the approach identifies new QTLs relevant to TB host response including specific genes that have defined roles in inflammation, wound healing, and other related responses. The associations of lesion necrosis with several chemokines and cytokines shown in Fig. 3 is an expected outcome, but one that is needed to establish relevant baselines for further identification of additional genes of interest. Importantly, it allows for weighting of the relationships to focus the downstream analysis. The identification of known QTLs and genes previously associated with disease outcomes such as S100a9 are important support the utility of the QTL mapping approach. The outcomes are potentially important and form the basis for several testable hypotheses of immune mediators that determine Mtb proliferation and lung pathology. The authors take an important step beyond predictive modeling to test candidate determinants of disease outcomes related to presence of the PWK/PhJ alleles or loss of S100a8 in mice challenged with Mtb. What is not clear in the machine learning approach is how the strategy was designed to identify for drivers versus outcomes. Mtb burden, tissue necrosis, and inflammation are already highly associated and Mtb burden is most often the driver of the inflammatory outcomes. The authors should clarify the assumptions used to design the analysis and caveats related to those assumption. Overall, these findings represent a substantial and successful effort to extricate predictive host signatures from among complex immune networks in a range of outbred animals that succumb across a fairly disparate timeline. There are several modifications needed (see below) to clarify the experimental sampling, timeline and rationale for selections. Minor editorial issues are noted.

**Part II – Major Issues: Key Experiments Required for Acceptance**

Reviewer #1: As the manuscipr arrived along previous reviews, the editors are aware that I have served as a reviewer during the first submission of the manuscript to PLoS Genetics, expressed my concerns and received adequate reponses from the authors. Since the manuscript underwent notable additional improvements, I have no major concerns.

Reviewer #2: None

Reviewer #3: The timelines and use of samples for particular analyses are very difficult to follow. An experimental timeline that indicates the samples harvested from particular mouse groups and the timepoint used for each particular analysis is needed. Presumably the ELISA data correlated with the pathology and other disease outcomes in Fig. 3 is 8 wk pi? It's unclear what timepoint the QTL mapping was done. Text suggests a time less than 250 days when animals met the criteria for euthanasia, which could be a progressor at ~20 days through a controller at 249 days? Overall, it seems strange that the progressor and controllers were not analyzed separately, so an explanation of the basis for pooling and handling the analysis is important.

Supplemental Fig. 3A is listed as pyknotic nuclear debris in a necrotizing granuloma. The authors should clarify how this was confirmed vs alveolar accumulation of mostly intact PMNs? Does accumulation alone meet criteria as necrotizing when the alveolar wall is intact? There are several very interesting granuloma features, as shown in Fig. 2 (e.g. Fig. 2E) that are mentioned in text but not illustrated. Since these lesions are similarly scored as necrotizing when developing the relationships, this distinction requires clarification.

Supplemental Figure 6 is confusingly incomplete. Survival data for mice with at least one PWK/PhJ allele indicate a survival disadvantage. Comparisons to the CAST/EiJ strain (low allelic expression) indicate Mtb lung burden is no different, yet only Mtb lung burden is shown for the S100A8 KO mice. It's unclear why histology, cytokines, or survival was not evaluated for S100A8 KO mice, given that these outcomes are important and key inputs to the model. If specimens are available, these analyses should be included.

**Part III – Minor Issues: Editorial and Data Presentation Modifications**

Reviewer #1: My minor concerns expressed previously were also met.

Reviewer #2: Considerations

1. Table 2 would be strengthened if it displayed a column with brief descriptions of the known gene functions. In several places the authors state “Table 2 summarizes the known annotations, allele effects, founder alleles, containing SNPs, and predicted effects of missense SNPs in the XXX gene on protein functions from publicly available databases” (eg lines 392-394). However, in the version I have, Table 2 does not describe gene function or SNP effects.

2. Lines 498-500. The authors report that the s100a8 heterozygote (+/-) and knockout (-/-) mutations in C57BL/6 mice showed minimal impact on ability to control Mtb lung CFUs. This contrasts with Scott et al. JCI 2020 (cited as ref. 91) who found that s100a8/a9 deficiency was significantly protective with lower lung CFU counts. It would be helpful for the authors to cite Scott et al. in this section of the text and to comment on the discordant results either in the Results or in the Discussion.

3. Line 480: This sentence refers to Fig. 9E, but I cannot find such a figure.

Reviewer #3: How many animals from each group are represented in many data sets is also difficult to assess. Figure legends do not consistently indicate this information. Also, is the data from all groups (progressors, etc) are pooled for QTL analysis? If so, how were comparisons among these groups made to inform the pooling criteria? Suppl Fig. 1 indicates 195 progressor and 145 controller mice. Gene expression is 98 mice, but how were they selected?

What was the basis of selection of analytes for the ELISA panel re: discovery? The panel is highly customized

Figure 5 overview is not a standalone figure, should be included within a figure as a visual, or moved to supplemental data.

Fig. 3. P values are somewhat difficult to see and numbers of replicates are not presented in the figure or figure legend. What timepoint post-infection is this? Are these Pearson correlation values shown significant? It's not clear if that is what the size of the elliptical shape that indicates strength of the association represents.

recommend use of greater and lesser as compared to higher and lower

Line 351: "remainder (of) lung"

Line 420: needs “a” in “Dots4 is new QTL”

Line 443 Dots6 is new QTL

Line 540: Mice this temporarily?

Line 625: we multiply

Line 646 punctuation in Dr

Why only look at Mtb burden with the S100a8 KO mice and not necrosis and inflammation, outcomes used to develop linkages?

PLOS authors have the option to publish the peer review history of their article (what does this mean?). If published, this will include your full peer review and any attached files.

Reviewer #1: **Yes: **Alexander Apt

Reviewer #2: No

Reviewer #3: No
---

## [Editor Report · Decision Letter 1]

17 Apr 2024

Dear Dr. Beamer,

We are pleased to inform you that your manuscript 'Systems genetics uncover new loci containing functional gene candidates in Mycobacterium tuberculosis-infected Diversity Outbred mice.' has been provisionally accepted for publication in PLOS Pathogens.

Best regards,

Padmini Salgame

Academic Editor

PLOS Pathogens

Michael Wessels

Section Editor

PLOS Pathogens

Michael Malim

Editor-in-Chief

PLOS Pathogens

orcid.org/0000-0002-7699-2064
---

## [Editor Report · Acceptance letter]

27 May 2024

Dear Dr. Beamer,

We are delighted to inform you that your manuscript, "Systems genetics uncover new loci containing functional gene candidates in Mycobacterium tuberculosis-infected Diversity Outbred mice.," has been formally accepted for publication in PLOS Pathogens.

Best regards,

Michael Malim

Editor-in-Chief

PLOS Pathogens

orcid.org/0000-0002-7699-2064